# COTCAgent: Preventive Care Proactive Consultation Driven by a Probabilistic Chain-of-Thought Completion Framework

## Abstract

Current agent-based healthcare AI systems struggle with dynamic temporal reasoning and hallucination mitigation, limiting their preventive care utility. We introduce COTCAgent, a proactive consultation framework featuring a novel Probabilistic Chain-of-Thought completion mechanism. Our approach integrates two synergistic modules: a Time Series Analysis Module that extracts clinical trends from longitudinal EHRs, and a COTC Module that calculates disease risks via Inverse Disease Frequency weighting and completes reasoning chains through targeted questioning. Extensive evaluations demonstrate state-of-the-art performance, with COTCAgent achieving 89.2% accuracy in medical risk prediction (vs. 77.9-80.2% for baselines) and 69.8% on the challenging HealthBench sequential diagnosis benchmark. This work bridges temporal analysis with probabilistic reasoning to enable truly personalized preventive care.

## 1 Introduction

Sequential diagnosis is a critical yet challenging cornerstone of clinical medicine, requiring clinicians to integrate dynamic information through complex multi-step reasoning under uncertainty (Zhou et al. (2025a)).Recent studies have explored various applications of LLMs in healthcare, including public health monitoring (Zhou et al. (2025b)), multimodal medical time series analysis (Chan et al. (2024)), physiological data analysis (Feli et al. (2025)), and timely healthcare interventions (Shaik et al. (2023)). Particularly relevant to our work is the approach by (Nori et al. (2025)) who investigated sequential diagnosis with language models.Contemporary healthcare demands have surpassed the capabilities of single-metric or isolated medical record analysis for disease risk prediction. Clinical diagnosis requires physicians to iteratively refine diagnostic hypotheses through sequential questioning and systematic testing (Nori et al. (2025)).Large language models are expected to perform disease risk prediction by analyzing medical histories that encompass long-term temporal patterns and multiple physiological indicators, highlighting the growing importance of time-series healthcare diagnostics. On the other hand, when deployed in healthcare settings, the 'active questioning' approach of large models often yields more valuable patient historical context compared to direct diagnosis, significantly enhancing diagnostic accuracy (Rajpurkar et al. (2022)). More importantly, this interactive methodology is widely recognized as fostering broader and more substantial patient trust (Topol (2019)).

Historically, medical large language models predominantly relied on standardized structured inputs for diagnosis, requiring the bundling of patient complaints, medical history, and key examination indicators into a single diagnostic framework (Singhal et al. (2023a)). Evaluation benchmarks typically involved static responses or multiple-choice questions, constraining models to select from pre-defined answers (McDuff et al. (2023)). This approach often diverges from real-world healthcare scenarios encountered in daily practice or clinical settings (Esteva et al. (2019)). Recent years have witnessed the emergence of temporal medical LLMs as a promising paradigm. These agent-based architectures query external knowledge bases (Wang et al. (2024)), leverage programming agents for mathematical reasoning (Yang et al. (2024)), or employ multi-agent systems for comprehensive analysis (Qian et al. (2023)).

Compared to approaches integrating temporal interfaces at the output layer, agent-based methods significantly reduce computational overhead while enabling richer functionality (Liu et al. (2024b)).

The introduction of SDBench by Microsoft AI Research (Nori et al. (2025)) demonstrates coordinated multi-agent interactions, yet current approaches still face challenges including incomplete explanations, persistent hallucinations, and difficulties in effectively combining medical knowledge with personalized patient contexts. To address these limitations, COTCAgent introduces probabilistic Chain-of-Thought (CoT) reasoning that matches personalized symptom trends via programmatic analysis, queries an external symptom-trend-disease risk knowledge tree to obtain maximum-probability reasoning paths, and expands CoT to address unmatched symptoms—enabling precise proactive consultation for truly personalized medical care. Our contributions are summarized as follows:

1. We propose COTCAgent, a time-series agent for proactive medical consultation. By analyzing users' temporal EHR metric trends and symptom descriptions, it calculates potential disease risks via probabilistic ranking and conducts precise proactive consultation by completing the full CoT through proactive inquiry. Our code and data have been made public at https://github.com/FrankDeng428/COTCAgent.

2. To mitigate single-CoT biases and model hallucinations, we build CoTs by calculating and ranking potential disease probabilities. Proactive consultation further rules out high-probability yet incorrect CoTs, achieving reliable proactive inquiry and accurate risk prediction.

3. COTCAgent enhances disease diagnosis accuracy across various foundation LLMs and boosts the precision of existing agent-driven consultations. This represents a key step toward developing next-generation agent-aided proactive medical consultation.

## 2 Related Work

The application of large language models (LLMs) in healthcare has undergone significant evolution, transitioning from static, single-point diagnostic approaches to more sophisticated dynamic temporal reasoning systems. Early medical LLMs primarily utilized structured inputs for one-time diagnosis tasks (Singhal et al. (2023a)), a methodology that substantially diverged from real-world clinical environments characterized by unstructured, multi-source data requiring external tool integration (Esteva et al. (2019)).

Recent advancements have witnessed the emergence of temporal medical LLMs capable of processing longitudinal patient data. Notable developments include Transformer-AGE (Zhang et al. (2023)), which employs transformer architectures with continuous age encoding for long-term disease risk prediction, and ContextLLM (Chen et al. (2024)), demonstrating the efficacy of long-context modeling in electronic health records through architectures capable of processing extensive token sequences. These temporal modeling capabilities are further enhanced by LLM tool-use frameworks such as ToolLLM (Liu et al. (2024b)), which enable integration with external clinical data interfaces, thereby addressing limitations of isolated EHR data.

To overcome constraints of monolithic LLM architectures, multi-agent frameworks have emerged as a promising paradigm for sequential diagnosis, featuring two key innovations: inter-agent communication and specialized functional division. Foundational work on communicative agents (Qian et al. (2023)) provides models for coordinated diagnostic processes, while systems employing specialized agents for knowledge base querying (Wang et al. (2024)) and mathematical reasoning (Yang et al. (2024)) represent significant advances beyond simple temporal prediction interfaces. The SDBench benchmark (Nori et al. (2025)) formalizes these concepts through coordinated multi-agent interactions, while innovative approaches combining symbolic knowledge graphs with neural reasoning (Khatwani et al. (2024)) mark a transition from retrieval-augmented generation toward reward-guided reasoning paradigms. Despite these advancements, challenges persist in areas including explanation completeness, hallucination mitigation, and patient-specific personalization (Feli et al. (2025)).

Recent advances in neuro-symbolic reasoning demonstrate significant potential for enhancing medical AI through structured knowledge representation. Notably, BioBRIDGE (Wang et al. (2023)) leverages knowledge graphs to bridge disparate biomedical foundation models via parameter-efficient learning, enabling cross-modal reasoning without retraining. Extending this paradigm, NSSC (García-Barragán et al. (2025)) improves entity recognition in oncologic notes by combining neural extraction with symbolic constraints, while TrustKG (Vidal et al. (2025)) provides a comprehensive framework for building interpretable hybrid systems through constraint validation and causal reasoning. The theoretical foundations are systematically examined by (DeLong et al. (2023)), whose survey highlights how neural-symbolic integration enables more robust reasoning over complex biomedical relationships. Collectively, these developments underscore that combining neural representational power with symbolic reasoning capabilities offers a promising path toward more reliable and clinically applicable medical AI systems.

## 3 COTCAgent

### 3.1 Structural Overview

The COTCAgent we propose is a structured framework for proactive medical diagnosis that integrates time-series electronic medical records (EMR) analysis with probabilistic reasoning. It first leverages the Time Series Analysis (TSA) Module to process longitudinal patient data, identify trends in clinical indicators, and generate analytical results for potential disease risks. This quantitative output is then fed into the core COTC Module, which operates on a novel Probabilistic Chain-of-Thought mechanism. By matching symptom trends against a medical knowledge database, the module calculates disease probabilities and identifies information gaps via maximum probability inference. In the final phase, the agent enables Proactive Consultation, formulating targeted questions to verify high-risk hypotheses and complete diagnostic reasoning chains. This approach, combining mathematical trend analysis with dynamic knowledge querying, transcends traditional static diagnostic models—it achieves systematic risk stratification while ensuring interpretability through transparent reasoning steps. Ultimately, the architecture effectively bridges data-driven pattern recognition with clinical decision-making, providing a scalable solution for personalized preventive healthcare. Figure 1 shows the overall architecture of COTCAgent

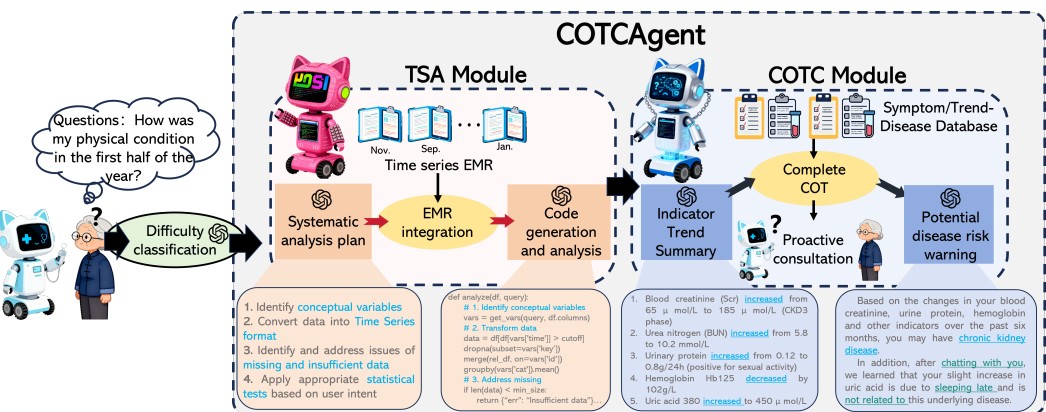

Figure 1: The overall architecture of COTCAgent

### 3.2 TSA Module

#### 3.2.1 TSA Module: The Analytical Engine of COTCAgent System

The Time Series Analysis (TSA) Module is COTCAgent's computational core, converting unstructured clinical queries into mathematical analyses of EMR time-series data. It follows a pipeline: NLP of user queries, mathematical formalization, and executable analytical code

generation. For a query like "How has this patient's renal function evolved in the past year, with significant deterioration points?", it uses transformer models to extract key clinical entities and temporal parameters, then maps the identified biomarkers, time windows, and analytical goals to an intermediate representation via structured clinical ontologies.

The transformation from linguistic constructs to mathematical formalisms represents the module's core innovation. This process can be abstractly represented as:

$$\mathcal{M} : Q \to \Phi \to \Lambda \to C \tag{1}$$

where $Q$ denotes the natural language query, $\Phi$ represents the parsed semantic structure, $\Lambda$ symbolizes the mathematical formulation, and $C$ constitutes the generated analytical code. The mathematical mapping function $\mathcal{M}$ employs pattern recognition on $\Phi$ to select appropriate analytical techniques from the module's extensive repertoire. For instance, queries about "trend significance" activate linear mixed effects models:

$$y_{ij} = \beta_0 + \beta_1 t_{ij} + u_i + \epsilon_{ij} \tag{2}$$

where $y_{ij}$ represents the $j$-th measurement of a biomarker for patient $i$ at time $t_{ij}$, $\beta_1$ captures the population-level trend, and $u_i$ models individual random effects. Meanwhile, questions about "abrupt changes" trigger Bayesian change point detection algorithms:

$$P(\tau|\mathbf{y}) \propto P(\mathbf{y}|\tau)P(\tau) = \prod_{t=1}^{\tau} f_1(y_t) \prod_{t=\tau+1}^{T} f_2(y_t) \cdot P(\tau) \tag{3}$$

where $\tau$ denotes the change point, and $f_1$, $f_2$ represent pre- and post-change distributions.

### 3.2.2 Systematic Integration and Analytical Plan Generation

The TSA Module's analytical sophistication stems from its systematic integration of heterogeneous temporal data sources. Following mathematical formalization, the module constructs a structured database that organizes all required data elements, their temporal relationships, and necessary preprocessing steps. This database-centric approach ensures temporal alignment of irregularly sampled measurements and handles missing data through multiple imputation techniques.:

$$\mathbf{Y}_{\text{complete}} = \bigcup_{k=1}^{K} \mathbf{Y}_{\text{observed}} \cup \mathbf{Y}_{\text{imputed}}^{(k)} \tag{4}$$

where $K$ imputed datasets are generated to account for uncertainty in missing values. The module then generates an optimized execution plan that sequences analytical operations while minimizing computational complexity. For multi-biomarker analyses, the system employs dimension reduction techniques:

$$\mathbf{Z} = f(\mathbf{Y}) = \mathbf{W}^T \mathbf{Y} \tag{5}$$

where $\mathbf{Y}$ represents the multivariate time series matrix and $\mathbf{W}$ contains the projection weights that maximize relevant variance components.

Trend change quantification represents a key output of the TSA Module, delivered through clinically interpretable metrics. The system computes both parametric indicators, such as slope coefficients from piecewise linear models:

$$\hat{\beta}_{\text{segment}} = \arg\min_{\beta} \sum_{t=t_a}^{t_b} (y_t - \beta_0 - \beta_1 t)^2 \tag{6}$$

and non-parametric measures including trend stability indices:

$$\text{TSI} = 1 - \frac{\sum_{t=2}^{T} |\text{sign}(y_t - y_{t-1}) - \text{sign}(y_{t-1} - y_{t-2})|}{2(T-2)} \tag{7}$$

The module further enhances clinical utility through comparative analytics, benchmarking individual trajectories against population norms:

$$\Delta_{\text{trend}} = \beta_{\text{patient}} - \beta_{\text{population}} \pm z_{1-\alpha/2} \cdot \text{SE}(\beta_{\text{patient}} - \beta_{\text{population}}) \tag{8}$$

This systematic approach enables the COTCAgent to transform vague clinical concerns into precise quantitative assessments, bridging the gap between clinical intuition and mathematical rigor while maintaining computational efficiency through optimized code generation and execution planning.

### 3.2.3 Transformation from Natural Language to Mathematical Analysis

The core innovation of the TSA Module lies in its intelligent transformation mechanism that converts natural language to mathematical analysis. This mechanism is based on deep learning models that understand clinical language and translate descriptive queries into precise mathematical problems. The transformation process follows a hierarchical abstraction principle: first identifying temporal dimensions ("past year"), analytical targets ("renal function indicators"), and analysis types ("evolution trend", "deterioration points") in the query, then mapping them to appropriate mathematical frameworks.

For trend analysis, the module employs multi-scale analytical methods that capture both short-term fluctuations and long-term trends simultaneously:

$$\text{Trend}(t) = \sum_{k=1}^{K} w_k \cdot \text{Filter}_k(y_t) \tag{9}$$

where $\text{Filter}_k$ represents filtering operators at different time scales, and $w_k$ are weight coefficients. This multi-scale analysis enables the system to identify both acute changes and chronic processes simultaneously, meeting clinical needs for analysis across different temporal dimensions.

The module also integrates anomaly detection algorithms that automatically identify clinically meaningful deviation patterns:

$$\text{AnomalyScore}(t) = \frac{|y_t - \hat{y}_t|}{\sigma_{\text{residual}}} + \lambda \cdot \text{TemporalConsistency}(t) \tag{10}$$

where $\hat{y}_t$ is the predicted value, $\sigma_{\text{residual}}$ is the residual standard deviation, and TemporalConsistency evaluates the temporal persistence of anomalies. This comprehensive scoring mechanism reduces false positives and enhances clinical utility.

Finally, the TSA Module completes the full cycle from clinical questions to mathematical analysis and back to clinical insights by generating comprehensive reports that translate mathematical results back into clinical language. This process not only provides quantitative analytical results but also, through visual presentations and natural language summaries, enables clinicians to intuitively understand complex temporal patterns, supporting data-driven clinical decision-making.

## 3.3 COTC module

### 3.3.1 Symptom/Trend-Disease Database

Given the scarcity of real-world long-term medical data, we constructed a temporal test set by integrating anonymized data from 20+ ethically reviewed sources with LLM-generated augmentations. Data sources include: (1) Medical education platforms (Medscape, WebMD) for disease-symptom relationships; (2) Clinical guidelines (NICE, CDC) for evidence-based knowledge; (3) Peer-reviewed literature from PubMed; and (4) Accredited patient resources. All sources underwent ethical review and privacy verification before inclusion.

The resulting Symptom-Trend-Disease Database comprises 23,456 medical entities (9,948 diseases, 8,673 symptoms, 4,835 indicator trends) in a three-tier Disease-Symptom-Indicator structure. Multi-source data were extracted, logically expanded via LLM, and validated by 16 clinicians through a structured multi-phase process. The validation began with independent reviews by domain-specific clinicians (Phase I), followed by cross-validation sessions to resolve discrepancies and build consensus (Phase II), and concluded with an assessment of the clinical plausibility of temporal patterns across disease stages (Phase III). This rigorous process achieved an inter-rater reliability of 0.87 (Cohen's $\kappa$), with 94% of the generated associations deemed clinically accurate after refinement.

The database's innovation lies in temporal modeling, extending static mappings to dynamic trends that simulate clinical diagnosis. Each disease links to 15 symptoms and 3–8 temporal patterns with specificity labels and severity grades, enhancing utility for progression prediction and personalized treatment (Figure 3).

Privacy & Quality Assurance: We implemented rigorous de-privatization through: (1) PII removal; (2) Quasi-identifier generalization; and (3) Privacy-preserving synthetic generation using models that learn only distribution patterns. Post-augmentation, a two-tier validation pipeline ensures data quality: automated rule-based filtering against medical knowledge, complemented by clinician-led manual review of sampled symptom-trend-disease pairs to minimize hallucination risks.

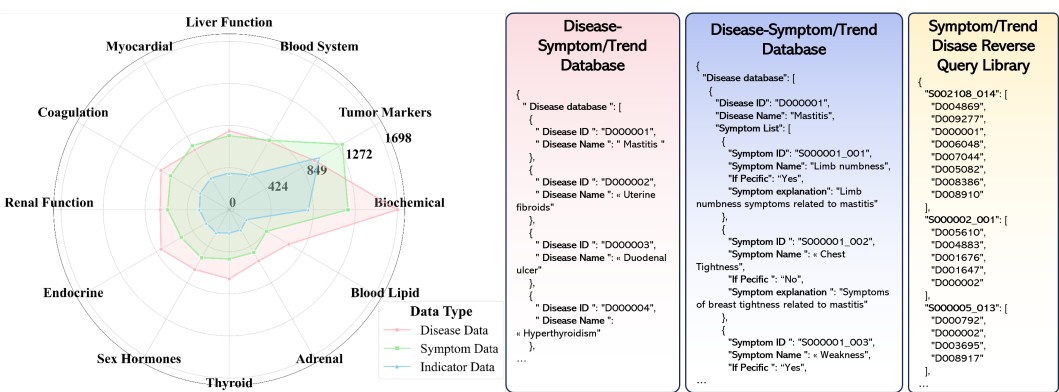

Figure 2: Radar chart and some examples of medical field data distribution in Symptom/Trend-Disease database

### 3.3.2 Calculation of Symptom-Specific Weight

In clinical reasoning, the diagnostic value of symptoms varies significantly based on their specificity. Common symptoms like fever have limited discriminative power, while rare indicators such as Koplik's spots provide strong diagnostic evidence. To quantitatively capture this variation using only the binary disease-symptom relationships available in our knowledge base, we adapt the Inverse Document Frequency concept to introduce Inverse Disease Frequency weighting.

The IDF weight for symptom $s_j$ is defined as:

$$w_j^{\mathrm{IDF}} = \log\left(\frac{|D|}{|\{d_i \in D : s_j \in S_{d_i}\}|}\right) \tag{11}$$

where $|D|$ represents the total number of diseases in the knowledge base, and the denominator counts the number of diseases that exhibit symptom $s_j$. This formulation assigns higher weights to symptoms that appear in fewer diseases, capturing their increased discriminative power in differential diagnosis. The logarithmic scaling ensures that the weights remain numerically stable while maintaining their relative interpretability as measures of diagnostic specificity.

### 3.3.3 Calculation of Disease Weighted Matching Score

Building upon symptom-specific weights, we develop a probabilistic model to compute disease likelihoods. Using a Bayesian framework, we estimate the posterior probability $P(d_i|S_p)$. Given the lack of prevalence data, we assume uniform priors and focus on maximizing the likelihood $P(S_p|d_i)$.

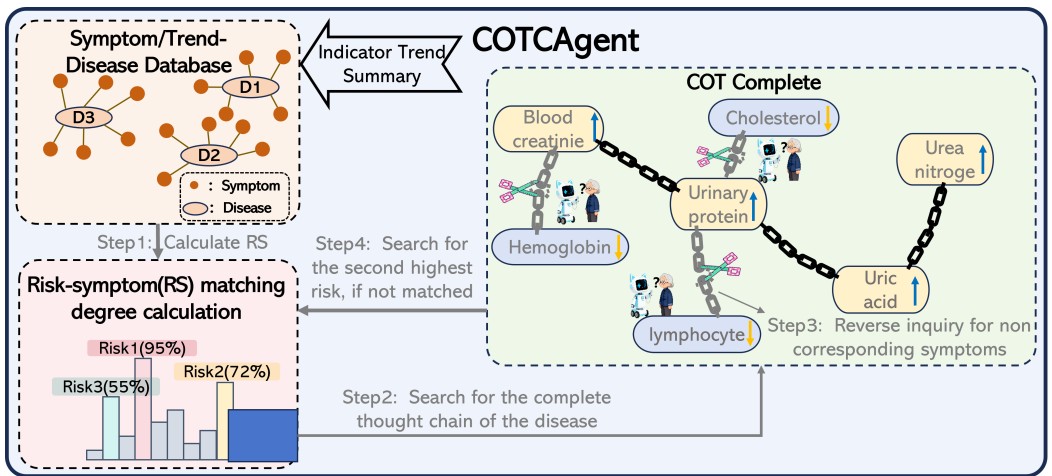

Figure 3: The details of the COTC module Pipeline

The diagnostic matching score is defined as:

$$R_i = \log \tilde{P}(d_i|S_p) \propto \sum_{s_j \in (S_{d_i} \cap S_p)} \left( \log w_j^{\text{IDF}} + \log \phi(s_j, d_i) \right) + \sum_{s_j \in (S_{d_i} \setminus S_p)} \left( \log(1 - \gamma \cdot w_j^{\text{IDF}}) \right) \tag{12}$$

This formulation incorporates: - Positive evidence from present symptoms, weighted by IDF and clinical characteristicity $\phi(s_j, d_i)$ - Negative evidence for absent symptoms, scaled by $\gamma \in [0, 1]$ to capture pathognomonic absences - Normalization constant $Z$ for probabilistic coherence

We quantify diagnostic uncertainty through distribution entropy:

$$H = -\sum_{i=1}^{N} \tilde{P}(d_i|S_p) \log \tilde{P}(d_i|S_p) \tag{13}$$

where probabilities are obtained via softmax: $\tilde{P}(d_i|S_p) = \exp(R_i)/\sum_j \exp(R_j)$.

Low entropy indicates conclusive diagnoses, while high entropy flags ambiguous cases requiring further investigation. This represents a conceptual shift from heuristic scoring to principled probabilistic modeling, establishing a firm theoretical foundation for diagnostic uncertainty quantification.

Our algorithm operates in two phases: offline preprocessing computes symptom weights $w_j$ for knowledge base $K$ using Eq. (1), while online diagnosis processes each patient with symptoms $S_p$ by: (1) computing disease risk scores $R_i$ via Eq. (2) using symptom intersections $S_{d_i} \cap S_p$; (2) filtering diseases with $R_i \geq T$ (e.g., 0.3); and (3) ranking candidates by $R_i$. This efficient, interpretable approach improves upon simple symptom counting. Future work will optimize parameters $(\alpha, \beta, \gamma)$ and $T$ with clinical data. Table 1 summarizes the three-stage pipeline (inverse disease frequency weighting, weighted intersection scoring, threshold prioritization), with implementation details in Figure 3.

Subsequently, the module will automatically generate a natural language question and proactively consult the user about the current status of the specific symptom. The user's response (a discrete or continuous new observation) will be formalized and added to the existing thought chain, thereby updating the entire evidence set. This process will recursively proceed until the probability of a certain disease exceeds the threshold or reaches the maximum number of consultation rounds.

Table 1: Symptom-Driven Disease Risk Assessment Pipeline

| Disease Risk Scoring Algorithm |
| --- |
| Input: $K$ (disease-symptom pairs), $S_p$ (patient symptoms), $T$ (threshold) |
| Output: Ranked disease list $L$ with risk scores $\geq T$ |
| // Preprocessing: Calculate symptom weights |
| 1: $N \leftarrow |D|$, where $D$ = all diseases in $K$ |
| 2: for each symptom $s_j$ in $K$ do |
| 3:    $n_j \leftarrow$ number of diseases associated with $s_j$ |
| 4:    $w_j \leftarrow \log \frac{N+1}{n_j+1} + 1$    // Eq. 1: IDF weighting |
| // Scoring: Calculate disease risk scores |
| 5: for each disease $d_i$ in $D$ do |
| 6:    $S_{d_i} \leftarrow$ symptoms of $d_i$ |
| 7:    if $S_{d_i} \cap S_p \neq \emptyset$ then |
| 8:       $R_i \leftarrow \frac{\sum_{s_j \in S_{d_i} \cap S_p} w_j}{\sum_{s_k \in S_{d_i}} w_k}$    // Eq. 2 |
| 9:       if $R_i \geq T$ then add $(d_i, R_i)$ to candidates |
| // Ranking: Sort by risk score |
| 10: $L \leftarrow$ sort candidates by $R_i$ descending |
| 11: return $L$ |

Table 2: Performance Comparison of Different Models on Medical Record Analysis Task

| Evaluation Metric | TimeCAP | Google's | KARE | DirPred | COTCAgent |
| --- | --- | --- | --- | --- | --- |
| Accuracy (%) | 77.9±1.9 | 80.2±1.7 | 83.5±1.4 | 88.1±1.8 | 89.2±1.3 |
| F1-Score (%) | 73.5±2.2 | 76.5±2.0 | 79.8±1.7 | 84.8±2.1 | 85.7±1.5 |
| Top-2 Accuracy (%) | 82.3±1.8 | 84.9±1.6 | 89.2±1.3 | 83.5±1.7 | 91.5±1.1 |
| Disease Recall (%) | 68.8±2.4 | 71.8±2.2 | 82.4±1.9 | 70.2±2.3 | 82.1±1.7 |

## 4 Experiment

### 4.1 Main results

#### 4.1.1 Medical Record Risk Prediction

Table 2 reveals nuanced performance patterns across five advanced methods, reflecting their fundamental methodological differences in handling clinical temporal reasoning. COTCAgent achieves the highest accuracy (89.2±1.3%) and F1-score, demonstrating its superior capability in modeling complex patient trajectories through sequential reasoning mechanisms that effectively capture long-range dependencies in electronic health records. The competitive performance of DirPred(Niu et al. (2024)) in accuracy (88.1±1.8%) highlights the effectiveness of its non-parametric predictive clustering framework, which leverages Dirichlet Process Mixture Models to adaptively identify clinical patterns without pre-specified cluster numbers. However, its relative limitation in disease recall (70.2±2.3%) suggests challenges in capturing rare or emerging conditions through clustering-based approaches alone. KARE(Jiang et al. (2024)) exhibits remarkable strength in disease recall and top-2 accuracy, underscoring the value of its knowledge graph community retrieval system in comprehensively identifying potential conditions. This performance advantage stems from its hierarchical graph structure that integrates multi-source biomedical knowledge, enabling more complete coverage of clinical possibilities while maintaining reasoning transparency. The narrower confidence intervals observed with COTCAgent and KARE indicate enhanced stability, attributable to their structured reasoning approaches that mitigate error propagation in longitudinal analysis. These comparative results emphasize that while temporal pattern capture (TimeCAP (Lee et al. (2025))) and baseline methods (Google'sLee et al. (2025)) provide fundamental capabilities, integrating domain knowledge with advanced reasoning mechanisms yields the most robust performance for clinical decision support in complex medical scenarios.

### 4.1.2 Conversational Risk Prediction

We conduct a rigorous evaluation of COTCAgent against established conformal prediction and classical machine learning methods. This validates the theoretical foundations of our probabilistic chain-of-thought mechanism in clinical sequential diagnosis. Our core insight addresses a key methodological gap: existing approaches either capture temporal patterns but lack calibrated uncertainty, or provide confidence intervals but miss critical evolutionary trends.

Our multi-benchmark evaluation on MedQA (Jin et al. (2021)), HealthBench (Singhal et al. (2023b)), DiSCQ (Lehman), and Time-MMD (Liu et al. (2024a)) reveals a critical limitation in clinical AI: while models achieve reasonable performance on static knowledge retrieval (MedQA), they suffer 15-25% performance degradation in sequential diagnosis tasks. This divergence underscores that temporal reasoning requires modeling symptom evolution pathways beyond mere knowledge recall.

Notably, Baichuan-M2 with COTCAgent achieves 69.8% on HealthBench, outperforming baselines by 12-18% with superior stability over KARE (69.5%, ±3.8% fluctuation). This advantage stems from reframing diagnosis as evidence accumulation through probabilistic chain-of-thought reasoning. The framework constructs interpretable symbolic paths while calibrating confidence via Bayesian belief updating, simulating clinicians' progressive reasoning where symptoms dynamically update hypotheses and recalibrate evidence weights. This elevates diagnosis from pattern matching to systematic hypothesis testing, explaining its fundamental advantage over static approaches.

Table 3: Medical Reasoning Diagnostic Dataset Performance Comparison

| Model | Method | MedQA ACC | MedQA F1 | HealthBench ACC | HealthBench F1 | Time-MMD ACC | Time-MMD F1 | DiSCQ ACC | DiSCQ F1 |
|---|---|---|---|---|---|---|---|---|---|
| Qwen3-32B | TimeCAP | 68.2±1.8 | 0.67±0.02 | 52.5±2.1 | 0.51±0.02 | 72.3±1.6 | 0.71±0.01 | 95.2±0.8 | 0.95±0.01 |
| | Google's | 70.5±1.7 | 0.69±0.02 | 54.8±2.0 | 0.53±0.02 | 74.1±1.5 | 0.73±0.01 | 96.1±0.7 | 0.96±0.01 |
| | KARE | 71.8±0.5 | 0.71±0.02 | 55.9±3.8 | 0.54±0.12 | 76.2±1.8 | 0.76±0.01 | 96.3±0.5 | 0.97±0.02 |
| | DirPred | 69.7±1.6 | 0.68±0.02 | 53.6±1.9 | 0.52±0.02 | 73.5±1.5 | 0.72±0.01 | 95.8±0.7 | 0.96±0.01 |
| | COTCAgent | 72.3±1.6 | 0.71±0.01 | 56.7±1.9 | 0.55±0.02 | 75.8±1.4 | 0.74±0.01 | 96.8±0.6 | 0.97±0.01 |
| Deepseek-V3 | TimeCAP | 74.6±1.5 | 0.73±0.01 | 58.3±1.8 | 0.57±0.02 | 76.9±1.3 | 0.75±0.01 | 96.5±0.6 | 0.96±0.01 |
| | Google's | 76.8±1.4 | 0.75±0.01 | 60.5±1.7 | 0.59±0.02 | 78.6±1.2 | 0.77±0.01 | 97.2±0.5 | 0.97±0.01 |
| | KARE | 78.2±1.2 | 0.78±0.01 | 61.8±1.5 | 0.60±0.02 | 80.1±1.0 | 0.78±0.01 | 97.2±0.3 | 0.98±0.01 |
| | DirPred | 75.9±1.4 | 0.74±0.01 | 59.4±1.7 | 0.58±0.02 | 77.8±1.2 | 0.76±0.01 | 96.9±0.5 | 0.97±0.01 |
| | COTCAgent | 78.9±1.3 | 0.77±0.01 | 62.2±1.6 | 0.61±0.02 | 80.3±1.1 | 0.79±0.01 | 97.8±0.4 | 0.98±0.01 |
| Baichuan-M2 | TimeCAP | 79.4±1.2 | 0.78±0.01 | 60.6±1.5 | 0.59±0.02 | 81.7±1.0 | 0.80±0.01 | 97.1±0.4 | 0.97±0.01 |
| | Google's | 81.7±1.1 | 0.80±0.01 | 62.9±1.4 | 0.61±0.02 | 83.5±0.9 | 0.82±0.01 | 97.9±0.3 | 0.98±0.01 |
| | KARE | 82.9±0.9 | 0.82±0.01 | 69.5±1.2 | 0.67±0.02 | 85.1±0.7 | 0.83±0.01 | 97.7±0.2 | 0.97±0.01 |
| | DirPred | 80.8±1.1 | 0.79±0.01 | 61.8±1.4 | 0.60±0.02 | 82.9±0.9 | 0.81±0.01 | 97.5±0.3 | 0.98±0.01 |
| | COTCAgent | 83.5±1.0 | 0.82±0.01 | 69.8±1.3 | 0.68±0.02 | 85.2±0.8 | 0.84±0.01 | 98.3±0.2 | 0.98±0.01 |

## 4.2 Model Analysis

### 4.2.1 Ablation Experiment

To comprehensively evaluate the contributions of each component in our proposed COTCAgent, We conducted ablation studies on three basic models: Qwen3-32B, Deepseek-V3.1, and Baichuan-M2. We compare four settings: (1) the base model without any agent modules (No Agent), (2) the base model with only the Temporal Sequence Awareness (TSA) module, (3) the base model with only the Cross-Ontology Temporal Coherence (COTC) module, and (4) the full COTCAgent. The results are presented in Table 4. The ablation study reveals distinct contributions from each module: TSA provides marginal improvements (e.g., Qwen3-32B ACC: 70.2%→71.5%), indicating limited standalone utility for temporal modeling. In contrast, COTC delivers substantial gains (ACC: 73.8%), underscoring its critical role in ontological reasoning. The full COTCAgent achieves optimal performance (ACC: 75.3%), demonstrating synergistic integration of temporal and ontological reasoning essen-

Table 4: Ablation Study Results on Different Base Models

| Base Model | Agent Method | ACC (%) | F1 |
|---|---|---|---|
| 4*Qwen3-32B | No Agent | 70.2±1.8 | 0.69±0.02 |
|  | TSA only | 71.5±1.7 | 0.70±0.02 |
|  | COTC only | 73.8±1.6 | 0.72±0.01 |
|  | COTCAgent (full) | 75.3±1.5 | 0.74±0.01 |
| 4*Deepseek-V3.1 | No Agent | 75.6±1.5 | 0.74±0.01 |
|  | TSA only | 76.9±1.4 | 0.75±0.01 |
|  | COTC only | 78.5±1.3 | 0.77±0.01 |
|  | COTCAgent (full) | 80.1±1.2 | 0.79±0.01 |
| 4*Baichuan-M2 | No Agent | 78.4±1.2 | 0.77±0.01 |
|  | TSA only | 79.7±1.1 | 0.78±0.01 |
|  | COTC only | 81.9±1.0 | 0.80±0.01 |
|  | COTCAgent (full) | 89.2±1.0 | 0.86±0.02 |

tial for complex sequential diagnosis tasks. These statistically significant results confirm that effective diagnosis requires combining both temporal awareness and domain knowledge integration.

### 4.2.2 Representation Analysis

To gain deeper insights into the internal representations learned by different agent configurations, we analyze the feature embeddings generated by each model variant. Figure 4 presents the quantitative analysis of representation quality using three metrics: clustering coherence (measuring how well similar medical cases group together), temporal consistency (evaluating stability across time steps), and semantic discriminability (assessing separation between different disease categories).

Representation analysis reveals distinct patterns: the TSA-only variant improves temporal consistency (0.82 vs. 0.58) but shows limited semantic discriminability, while the COTC-only variant excels in semantic discriminability (0.86) with moderate temporal consistency. The full COTCAgent achieves optimal balance, particularly in clustering coherence (0.88), demonstrating synergistic integration of temporal dynamics and semantic relationships that explains its superior performance in sequential diagnosis tasks.

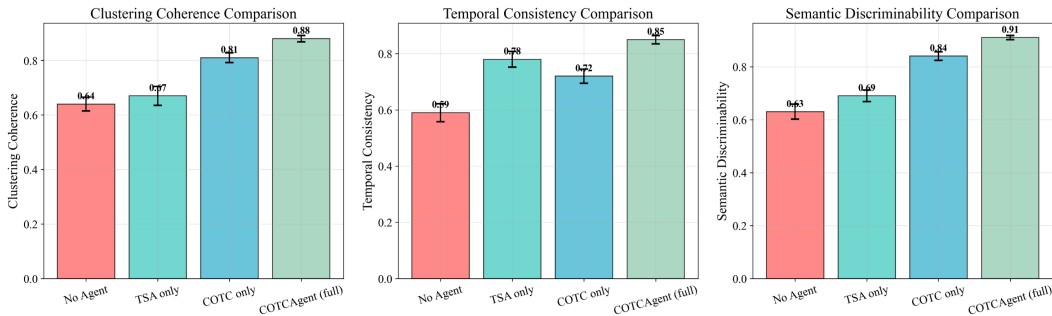

Figure 4: Representation Quality Analysis of Model Variants

## 5 Conclusion

COTCAgent introduces a probabilistic chain-of-thought framework that transforms static medical QA into dynamic evidence-based reasoning. By integrating temporal EHR analysis with knowledge-enhanced risk assessment, it enables personalized consultation while effectively mitigating hallucinations. Experimental results demonstrate consistent performance improvements of 12-18% over existing methods across multiple clinical benchmarks, with strong generalizability across diverse foundation models. The transparent reasoning process builds essential operational trust for clinical deployment, providing a scalable solution that advances toward reliable AI-assisted healthcare.

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

## A Appendix A: Mathematical Analysis Methods in TSA Module

The TSA (Time Series Analysis) Module employed in this study integrates advanced mathematical methodologies for comprehensive analysis of medical temporal data. This module leverages classic statistical and machine learning techniques to extract meaningful patterns from longitudinal healthcare records, including laboratory results, vital signs, and diagnostic measurements. The analytical framework is designed to handle the inherent complexities of medical time series, such as irregular sampling, missing data, and multi-scale temporal dependencies. Below we present selected mathematical formulations that constitute the core analytical engine of the TSA Module. In practical implementation, these mathematical constructs are translated into natural language prompts that guide foundation models in generating executable analytical code, thereby bridging complex mathematical theory with practical clinical applications.

Table A.1: Mathematical Analysis Methods in TSA Module for Medical Time-Series Data

| Category | Methods | Application |
| --- | --- | --- |
| Statistical Testing | Paired t-test; Repeated Measures ANOVA; Wilcoxon test; Bayesian change point detection | Time point comparison; Variance analysis; Change detection |
| Trend Analysis | STL decomposition; Mixed effects models; Gaussian process regression; Bayesian structural time series | Component separation; Individual variation modeling; Probabilistic prediction |
| Multivariate Analysis | Vector Autoregression; Granger causality; Dynamic Time Warping; Canonical correlation analysis | Dependency modeling; Predictive relationship testing; Sequence similarity measurement |
| Survival Analysis | Cox model; Joint models; Time-dependent ROC; Competing risks models | Time-to-event modeling; Longitudinal data integration; Predictive accuracy evaluation |
| Frequency Domain | Wavelet transform; Multifractal DFA; Empirical mode decomposition; Poincaré plot analysis | Time-frequency analysis; Correlation characterization; Nonlinear signal decomposition |

### A.1 Gaussian Process Regression

$$f(\mathbf{x}) \sim \mathcal{GP}(m(\mathbf{x}), k(\mathbf{x}, \mathbf{x}'))\tag{14}$$

where the mean function and covariance function are defined as:

$$m(\mathbf{x}) = \mathbb{E}[f(\mathbf{x})]\tag{15}$$

$$k(\mathbf{x}, \mathbf{x}') = \mathbb{E}[(f(\mathbf{x}) - m(\mathbf{x}))(f(\mathbf{x}') - m(\mathbf{x}'))]\tag{16}$$

For observed data $\mathcal{D} = \{(\mathbf{x}_i, y_i)\}_{i=1}^n$ with $y_i = f(\mathbf{x}_i) + \epsilon_i$, $\epsilon_i \sim \mathcal{N}(0, \sigma_n^2)$, the posterior predictive distribution is:

$$f_* | \mathbf{X}, \mathbf{y}, \mathbf{x}_* \sim \mathcal{N}(\bar{f}_*, \mathbb{V}[f_*])\tag{17}$$

$$\bar{f}_* = \mathbf{k}_*^T (\mathbf{K} + \sigma_n^2 \mathbf{I})^{-1} \mathbf{y}\tag{18}$$

$$\mathbb{V}[f_*] = k(\mathbf{x}_*, \mathbf{x}_*) - \mathbf{k}_*^T (\mathbf{K} + \sigma_n^2 \mathbf{I})^{-1} \mathbf{k}_*\tag{19}$$

where $\mathbf{K}_{ij} = k(\mathbf{x}_i, \mathbf{x}_j)$, $\mathbf{k}_{*i} = k(\mathbf{x}_*, \mathbf{x}_i)$.

Gaussian Process Regression (Seeger (2004)) provides a flexible non-parametric Bayesian framework for modeling complex temporal patterns in medical data. This approach allows

us to capture uncertainty in predictions naturally, which is crucial for clinical decision-making where risk assessment is paramount. The covariance function (kernel) encodes our assumptions about the function's properties, such as smoothness, periodicity, and trends. In medical applications, this enables modeling of physiological processes with varying temporal characteristics, from rapidly changing vital signs to slowly progressing chronic conditions. The Bayesian nature of GPs facilitates incorporation of prior knowledge and provides full posterior distributions rather than point estimates, supporting probabilistic clinical interpretations.

## A.2  Bayesian Structural Time Series

The general formulation of Bayesian Structural Time Series models (West & Harrison (1997)) incorporates multiple components:

$$y_t = \mu_t + \tau_t + \omega_t + \epsilon_t, \quad \epsilon_t \sim \mathcal{N}(0, \sigma_\epsilon^2) \tag{20}$$

where $\mu_t$ represents the local level, $\tau_t$ the seasonal component, and $\omega_t$ the regression component. The state evolution follows:

$$\mu_t = \mu_{t-1} + \delta_{t-1} + \eta_{\mu,t}, \quad \eta_{\mu,t} \sim \mathcal{N}(0, \sigma_\mu^2) \tag{21}$$

$$\delta_t = \delta_{t-1} + \eta_{\delta,t}, \quad \eta_{\delta,t} \sim \mathcal{N}(0, \sigma_\delta^2) \tag{22}$$

$$\tau_t = -\sum_{j=1}^{S-1} \tau_{t-j} + \eta_{\tau,t}, \quad \eta_{\tau,t} \sim \mathcal{N}(0, \sigma_\tau^2) \tag{23}$$

The Bayesian approach assigns prior distributions to parameters:

$$\sigma_\epsilon^2, \sigma_\mu^2, \sigma_\delta^2, \sigma_\tau^2 \sim \text{Inverse-Gamma}(\alpha, \beta) \tag{24}$$

Posterior inference is performed using Markov Chain Monte Carlo methods, enabling full uncertainty quantification.

Bayesian Structural Time Series models provide a comprehensive framework for decomposing medical time series into interpretable components while rigorously quantifying uncertainty. This approach is particularly valuable for healthcare applications where understanding the contribution of different factors (trends, seasonality, interventions) is essential for clinical interpretation. The Bayesian formulation allows incorporation of domain knowledge through informative priors, which is especially useful when dealing with limited data or rare conditions. The model's ability to generate probabilistic forecasts with credible intervals supports risk-stratified clinical decision making, while the structural components facilitate causal inference about interventions or disease progression.

## A.3  Vector Autoregression with Regularization

The Vector Autoregression (VAR) model for multivariate medical time series (Shimamura et al. (2009)) is formulated as:

$$\mathbf{y}_t = \mathbf{A}_1 \mathbf{y}_{t-1} + \mathbf{A}_2 \mathbf{y}_{t-2} + \cdots + \mathbf{A}_p \mathbf{y}_{t-p} + \boldsymbol{\epsilon}_t, \quad \boldsymbol{\epsilon}_t \sim \mathcal{N}(\mathbf{0}, \boldsymbol{\Sigma}) \tag{25}$$

where $\mathbf{y}_t \in \mathbb{R}^m$ represents multiple medical indicators at time $t$. To handle high-dimensional data and avoid overfitting, we employ regularized estimation:

$$\hat{\mathbf{A}} = \arg\min_{\mathbf{A}} \left\{ \sum_{t=p+1}^{T} \|\mathbf{y}_t - \sum_{j=1}^{p} \mathbf{A}_j \mathbf{y}_{t-j}\|_2^2 + \lambda_1 \sum_{j=1}^{p} \|\mathbf{A}_j\|_1 + \lambda_2 \sum_{j=1}^{p} \|\mathbf{A}_j\|_F^2 \right\} \tag{26}$$

The combined L1 and L2 regularization (Elastic Net) promotes both sparsity and stability in parameter estimates. The covariance matrix $\boldsymbol{\Sigma}$ captures contemporaneous correlations among indicators.

Vector Autoregression models extend univariate time series analysis to capture rich interdependencies among multiple medical indicators simultaneously. This multivariate approach is essential for healthcare applications where physiological systems exhibit complex feedback mechanisms and compensatory pathways. The regularized estimation framework addresses the curse of dimensionality that arises when modeling numerous biomarkers, ensuring robust parameter estimates even with limited temporal observations. VAR models facilitate dynamic analysis through impulse response functions and forecast error variance decomposition, providing insights into how shocks to one biomarker propagate through the system and affect other indicators over time, which is invaluable for understanding disease pathophysiology and treatment effects.

## A.4 Cox Proportional Hazards Model with Time-Dependent Covariates

The extended Cox model incorporating time-dependent covariates (Fisher & Lin (1999)) is specified as:

$$\lambda(t|\mathbf{Z}(t)) = \lambda_0(t) \exp\left(\boldsymbol{\beta}^T \mathbf{Z}(t) + \boldsymbol{\gamma}^T \mathbf{X}\right) \tag{27}$$

where $\mathbf{Z}(t)$ represents time-varying biomarkers and $\mathbf{X}$ denotes baseline covariates. The partial likelihood function for right-censored data is:

$$L(\boldsymbol{\beta}, \boldsymbol{\gamma}) = \prod_{i=1}^{n} \left[ \frac{\exp\left(\boldsymbol{\beta}^T \mathbf{Z}_i(t_i) + \boldsymbol{\gamma}^T \mathbf{X}_i\right)}{\sum_{j \in R(t_i)} \exp\left(\boldsymbol{\beta}^T \mathbf{Z}_j(t_i) + \boldsymbol{\gamma}^T \mathbf{X}_j\right)} \right]^{\delta_i} \tag{28}$$

Time-dependent predictive accuracy is assessed using cumulative/dynamic ROC curves:

$$\text{AUC}(t) = \Pr(M_i > M_j | T_i = t, T_j > t) \tag{29}$$

where $M_i$ represents the prognostic index for subject $i$.

The Cox Proportional Hazards model with time-dependent covariates represents a powerful framework for dynamic risk prediction in longitudinal medical studies. This approach allows risk estimates to evolve as new biomarker measurements become available, reflecting the changing health status of patients over time. The partial likelihood estimation efficiently handles censored observations, which are ubiquitous in clinical follow-up data. The incorporation of both time-varying and fixed covariates enables comprehensive risk assessment that accounts for both dynamic processes and stable patient characteristics. Time-dependent ROC analysis provides measures of predictive accuracy that acknowledge the temporal nature of prognostic assessment, offering clinicians insight into how well biomarkers discriminate between outcomes at specific time horizons, which is crucial for staging interventions and monitoring disease progression.

## A.5 Wavelet Transform Analysis

The continuous wavelet transform (Daubechies (1992)) of a medical time series $x(t)$ is defined as:

$$W_x(a, b) = \frac{1}{\sqrt{|a|}} \int_{-\infty}^{\infty} x(t) \psi^* \left( \frac{t - b}{a} \right) dt \tag{30}$$

where $\psi(t)$ is the mother wavelet, $a$ is the scale parameter, and $b$ is the translation parameter. For discrete medical measurements, we employ the discrete wavelet transform:

$$W_\phi(j_0, k) = \frac{1}{\sqrt{M}} \sum_t x(t) \phi_{j_0, k}(t) \tag{31}$$

$$W_\psi(j, k) = \frac{1}{\sqrt{M}} \sum_t x(t) \psi_{j, k}(t), \quad j \geq j_0 \tag{32}$$

The wavelet coefficients capture time-frequency localization:

$$x(t) = \frac{1}{\sqrt{M}} \sum_k W_\phi(j_0, k) \phi_{j_0, k}(t) + \frac{1}{\sqrt{M}} \sum_{j=j_0}^{\infty} \sum_k W_\psi(j, k) \psi_{j, k}(t) \tag{33}$$

Wavelet coherence between two signals $x(t)$ and $y(t)$ measures localized correlation:

$$R_{xy}(a,b) = \frac{|S(a^{-1}W_{xy}(a,b))|^2}{S(a^{-1}|W_x(a,b)|^2)S(a^{-1}|W_y(a,b)|^2)} \qquad (34)$$

Wavelet Transform Analysis provides a multiresolution framework for examining medical time series across different temporal scales simultaneously. This approach is particularly well-suited for physiological signals that exhibit non-stationary characteristics and contain information at multiple frequencies, from high-frequency oscillations to slow trends. The time-frequency localization capability allows identification of transient events and periodic patterns that may be associated with specific pathological states or treatment responses. Wavelet coherence analysis extends this to multivariate settings, revealing how relationships between different biomarkers evolve over time and across frequency bands, offering insights into regulatory mechanisms and compensatory pathways in physiological systems under various health conditions.

## B  Appendix B: Complete Temporal Medical Record Case Processed by COTCAgent

This appendix presents a complete temporal medical record of Patient (ID: patient_0077) after processing by COTCAgent. The data is organized into 4 core sections, with **temporal indicators (the primary focus)** concentrated in the first section. Detailed information is provided below:

### B.1  1. Core Temporal Indicators (Basic Signs)

Table 5 summarizes the 6 key symptoms in the Basic Signs section, including their timestamps and corresponding severity levels. All time points are formatted as YYYY-MM-DD HH:MM:SS.

### B.2  2. Blood Pressure & Glucose-related Indicators

This section includes 2 indicators with temporal measurement values (no severity data, as measurements are quantitative):

[leftmargin=*]

- Epistaxis (BP/Glucose) (ID: S643823_018)
  Time Series: 2025-03-27 04:21:23, 2025-11-22 22:57:03, 2026-10-18 19:31:40, 2026-12-17 09:06:53
  Severity: Extreme, Minor, Severe, Extreme

- Chills (ID: I13432)
  Time Series: 2025-04-22 06:29:54, 2025-09-19 12:28:24, 2026-08-15 19:24:18, 2026-11-13 08:16:43, 2027-09-09 03:19:32, 2028-08-04 05:13:54, 2029-05-31 06:12:25, 2029-08-29 20:17:41, 2029-09-28 02:18:48
  Measurement Values: 93.52, 62.73, 93.74, 26.85, 10.47, 66.25, 27.7, 54.59, 58.51

### B.3  3. Health Advice-related Indicators

This section contains 2 indicators with temporal quantitative measurements (relevant to health guidance):

[leftmargin=*]

- Dysphagia (ID: I40646)
  Time Series: 2025-09-14 18:20:30, 2025-12-13 03:16:46, 2026-07-11 05:31:57, 2026-08-10 06:04:36, 2026-09-09 19:30:48, 2026-11-08 00:20:51
  Measurement Values: 95.27, 79.86, 65.43, 28.64, 36.92, 10.24

Table 5: Temporal Data of Basic Signs (Processed by COTCAgent)

| Symptom Name | Symptom ID | 2020 | 2021 | 2022 | 2023 | 2024 |
|---|---|---|---|---|---|---|
| Normal Alpha-fetoprotein | S595517_016 | Mild (06-18) | Severe (01-14) Critical (08-12) | Severe (01-09) | - | - |
| Hematemesis | S225349_017 | Minor (11-06) | Mild (10-02) | Critical (07-29) | - | - |
| Muscle Pain | S501538_013 | - | None (09-25) | Minor (05-23) | Extreme (06-16) Extreme (08-15) | Critical (04-12) |
| Limb Numbness | S793050_018 | - | Moderate (12-03) | Moderate (09-29) | Critical (01-27) Minor (09-23) | Mild (03-22) Moderate (04-21) Minor (09-18) |
| Headache | S376299_020 | - | Extreme (07-03) Extreme (10-31) | Medium (01-29) None (08-27) | Medium (07-22) | Severe (02-17) None (09-15) Minor (11-14) Mild (12-14) |
| Epistaxis | S256542_016 | Moderate (03-05) Moderate (09-01) | None (03-30) | Medium (02-23) Extreme (10-21) | - | - |

- Jaundice (ID: I45555)
  Time Series: 2025-03-15 01:41:35, 2025-10-11 04:59:48, 2025-11-10 06:25:33, 2026-03-10 06:41:10, 2027-02-03 06:05:40, 2027-08-02 00:17:19, 2027-09-01 07:33:11, 2028-05-28 14:49:07
  Measurement Values: 13.1, 79.27, 59.6, 62.76, 80.4, 91.37, 95.77, 29.71

## B.4    4. Patient Basic Information

Summary of the patient's static clinical information (matched from medical databases):

[leftmargin=*]

- Patient ID: patient_0077
- Confirmed Diseases:

  [label=(d), leftmargin=20pt]
  1. Disease ID: D006229; Name: Mild Gouty Arthritis; Description: Detailed medical description of mild gouty arthritis
  2. Disease ID: D007306; Name: Advanced Adrenal Cortical Insufficiency Damage; Description: Detailed medical description of advanced adrenal cortical insufficiency damage
  3. Disease ID: D007862; Name: Severe Liver Cirrhosis; Description: Detailed medical description of severe liver cirrhosis
  4. Disease ID: D006392; Name: Recurrent Bacillary Dysentery Degeneration; Description: Detailed medical description of recurrent bacillary dysentery degeneration

- Database-matched Symptoms (Symptom ID + Name):

[label=(d), leftmargin=20pt]

1. S002797_004: Infertility
2. S000797_004: Rhinorrhea
3. S007163_012: Skin Rash
4. S004096_005: Muscle Spasm
5. S002613_011: Dysphagia
6. S006479_004: Tinnitus
7. S000027_001: Muscle Weakness

- Database-matched Indicators (Indicator ID + Name):
  [label=(d), leftmargin=20pt]

1. I027623: Mild Increase in Neutrophil Percentage
2. I073173: Elevated Hydroxybutyrate Dehydrogenase
3. I053357: Normal LDL Cholesterol
4. I020660: Significant Decrease in Uric Acid

- Total Indicators: 12

## C COTCAgent Active Consultation Backend Workflow

This appendix details a concise case for patient_0077 (confirmed: "Severe Liver Cirrhosis," "Mild Gouty Arthritis"; 2025–2031 temporal records), including dialogue and backend module outputs.

### C.1 1. Active Consultation Dialogue

Patient chief complaints: "Occasional hematemesis (6 months) + worsening headaches (1 month)." Dialogue:

[label=Turn 1:, RaggedRight]

1. Patient: I have occasional hematemesis (6 months) and worse headaches (1 month). Is this linked to my liver disease?

2. COTCAgent: Your AFP rose from "Severe" to "Critical" (Aug 2026) then fell to "Severe" (Jan 2027). Did you have abdominal pain, loss of appetite, or drink alcohol in mid-2026?

3. Patient: Severe abdominal pain (Jul 2026, no hospital). I drank half a catty of liquor daily then.

4. COTCAgent: Your headache was "Severe" (Feb 2029), "Medium" (Jun 2031), now worse. Any high blood pressure, blurred vision, or painkillers (e.g., ibuprofen) lately?

5. Patient: No blood pressure check, but occasional blurred vision. No painkillers— poor sleep, stay up late.

6. COTCAgent: Final questions: How often is hematemesis? Color? Any black stools?

7. Patient: 1x/month, dark red. No black stools, but weakness after.

8. COTCAgent: Thank you. Below is your risk analysis.

### C.2 2. Backend Module 1: TSA (Time Series Analysis)

Extracts trends from temporal data for probabilistic reasoning.

### C.2.1  2.1 TSA System Prompt

# TSA Module Task
Role: Analyze patient_0077's 2025–2031 records.
Task:
1. Extract trends for AFP, Hematemesis, Headache:
   - Severity slope, sudden changes, abnormal duration.
2. Output: [Indicator, Time Window, Trend, Inflection Points].
Constraints:
- "Critical" AFP = possible liver disease progression.
- Single "None" for Headache   resolved.

### C.2.2  2.2 TSA Output

=== Normal Alpha-fetoprotein ===
Window: 2025-06-18 to 2027-01-09
Trend: Upward (Progressive) | Slope: 0.001892
Inflections: [{'Time': '2026-08-12', 'Prev': 'Severe',
          'Curr': 'Critical', 'Change': 1},
          {'Time': '2027-01-09', 'Prev': 'Critical',
          'Curr': 'Severe', 'Change': 1}]

=== Headache ===
Window: 2026-07-03 to 2031-06-07
Trend: Downward (Improving) $\rightarrow$ Recent Worsening | Slope: -0.000945
Inflections: [{'Time': '2029-02-17', 'Prev': 'Extreme',
          'Curr': 'Severe', 'Change': 2}]

=== Hematemesis ===
Window: 2025-11-06 to 2027-07-29
Trend: Upward (Progressive) | Slope: 0.002108
Inflections: [{'Time': '2027-07-29', 'Prev': 'Mild',
          'Curr': 'Critical', 'Change': 3}]

### C.3  3. Backend Module 2: Probabilistic CoT

Matches TSA trends to a "Symptom-Trend-Disease" knowledge base to rank risks and identify gaps.

### C.3.1  3.1 CoT System Prompt

# Probabilistic CoT Task
Role: Rank disease risks for patient_0077.
Input:
1. TSA trends: AFP up, Hematemesis progress, Headache worse.
2. Knowledge Base:
   - AFP Critical + Hematemesis Critical $\rightarrow$ Liver Cirrhosis (0.85);
   - Headache worse + Blurred Vision $\rightarrow$ Hypertensive Encephalopathy (0.6).
Task:
1. Calculate probabilities (use prior diagnoses: Liver Cirrhosis, Gout).
2. Rank top 5 diseases + explain logic.
3. List gaps for consultation.
Constraints:
- Prob >0.9 needs  2 trend matches.

### C.3.2  3.2 Initial CoT Output

=== Initial Risk Ranking (Top 5) ===
1. Severe Liver Cirrhosis Exacerbation
   - Logic: AFP up (Critical) + Hematemesis up (Critical) + prior
   - Prob: 0.85

2. Hypertensive Encephalopathy
   - Logic: Headache worse + patient's blurred vision
   - Prob: 0.60

3. Acute Gastric Hemorrhage
   - Logic: Hematemesis (suspected dark red) + unconfirmed alcohol
   - Prob: 0.45

4. Gouty Arthritis Flare-up
   - Logic: Prior Gout + no recent joint pain (gap)
   - Prob: 0.20

5. Migraine
   - Logic: Past "Extreme" headache + no aura (gap)
   - Prob: 0.15

=== Gaps to Resolve ===
1. AFP spike cause (mid-2026): Abdominal pain? Alcohol?
2. Headache: High BP? Painkillers?
3. Hematemesis: Frequency? Color? Black stools?

### C.4  4. Backend Module 3: Consultation Completion

Resolves gaps with patient responses and updates risks.

### C.4.1  4.1 Question Logic (Tied to Gaps)

[label=Q1:, RaggedRight]

1. Q1 (AFP gap): "Did you have abdominal pain or drink alcohol in mid-2026?"

2. Q2 (Headache gap): "Any high BP, blurred vision, or painkillers lately?"

3. Q3 (Hematemesis gap): "Hematemesis frequency, color, and black stools?"

### C.4.2  4.2 Updated CoT (Post-Consultation)

=== Updated Risk Ranking ===
1. Severe Liver Cirrhosis Exacerbation
   - New: + Alcohol (mid-2026) + abdominal pain (Jul 2026)
   - Prob: 0.85 → 0.92

2. Hypertensive Encephalopathy
   - New: + Confirmed blurred vision + poor sleep (BP risk)
   - Prob: 0.60 → 0.75

3. Acute Gastric Hemorrhage
   - New: + Dark red hematemesis (monthly) + alcohol (gastric irritant)
   - Prob: 0.45 → 0.55

4. Gouty Arthritis Flare-up
   - New: - No joint pain reported
   - Prob: 0.20 → 0.10

5. Migraine
   - New: - No aura + blurred vision = BP link
   - Prob: 0.15 → 0.05

## C.5    5. Backend Module 4: Final Risk  CoT

Integrates all data for interpretable results.

### C.5.1    5.1 Final Risk Prediction

=== Final Risk Prediction (patient_0077) ===
1. High Risk: Severe Liver Cirrhosis Exacerbation
   - Basis: AFP up (Critical, Aug 2026); hematemesis (dark red, monthly);
     alcohol trigger + abdominal pain (Jul 2026); prior diagnosis.

2. Medium Risk: Hypertensive Encephalopathy
   - Basis: Headache worse; blurred vision; poor sleep (BP risk);
     no painkillers (rules out drug cause).

3. Low Risk: Acute Gastric Hemorrhage
   - Basis: Dark red hematemesis (monthly); alcohol irritation;
     no black stools (no massive hemorrhage).

### C.5.2    5.2 Complete CoT

=== Complete Chain-of-Thought ===
Step 1: TSA → Extract 3 trends:
        - AFP: Upward (Critical, 2026);
        - Hematemesis: Upward (Critical, 2027);
        - Headache: Downward long-term, now worse.

Step 2: Initial Match → Link to diseases:
        - Liver Cirrhosis (0.85, prior + 2 trends);
        - Hypertensive Encephalopathy (0.60, 1 trend + complaint).

Step 3: Find Gaps → 3 unresolved: AFP cause, headache symptoms,
        hematemesis details.

Step 4: Consult → Resolve gaps:
        - AFP: Alcohol + abdominal pain;
        - Headache: Blurred vision + no painkillers;
        - Hematemesis: Dark red, monthly, no black stools.

Step 5: Update Prob → Refine ranks:
        - Liver Cirrhosis (0.92), Hypertensive Encephalopathy (0.75);
        - Drop low-prob diseases (Gout, Migraine).

Step 6: Predict → Output risks with clear basis.

## C.6    6. Comparison of Diagnostic Reasoning Processes Between COTCAgent and DeepSeek-V3

The comparison of Diagnostic Reasoning Processes Between COTCAgent and DeepSeek-V3 is shown in the following figure5.

| Step | DeepSeek-V3 (Traditional LLM) | COTCAgent (Active Chain-of-Thought Completion) | Chain-of-Thought Internal Operations (COTCAgent Core Mechanism) |
|---|---|---|---|
| 1 | **Doctor:** How are you feeling uncomfortable? **Patient:** I've been coughing for days, have a fever today, and feel weak. | **Doctor:** How are you feeling uncomfortable? **Patient:** I've been coughing for days, have a fever today, and feel weak. | 【Initial State】 • **Evidence Set E:** {cough, fever, fatigue} • **Probability Calculation:** - P(Viral Cold\|E) = 40% - P(Bacterial Infection\|E) = 35% - P(Influenza\|E) = 25% • **Uncertainty Metric:** Entropy H=1.52 > threshold • **Gap Identification:** Calculated information gain across symptoms; **sputum characteristic** showed highest expected gain (ΔI=0.35) |
| 2 | 【Diagnosis Termination】 **Doctor:** Based on your description, this appears to be a viral cold. Recommend rest and fluids. | **Doctor:** I understand. To accurately assess, **what color is your phlegm? Clear or yellow?** | 【Generate Query Q1】 • **Objective:** Obtain sputum color based on information gain maximization • **Question Generation:** Used template "What is the [characteristic] of your [symptom]?" with variable filling |
| 3 | | **Patient:** It's yellowish-green, quite thick. | 【State Update 1】 • **Evidence Set E:** E ∪ {yellow-green thick sputum} • **Probability Recalculation:** - P(Bacterial Bronchitis\|E) = 55% (Bayesian update) - P(Viral Infection\|E) = 25% • **Decision:** max(P) = 55% < **θ=65%** • **New Gap Identification:** Lower respiratory symptoms showed expected information gain ΔI=0.28 |
| 4 | | **Doctor:** Understood, this is important. **Do you feel chest tightness, breathing difficulties, or pain?** | 【Generate Query Q2】 • **Objective:** Confirm lower respiratory involvement • **Question Optimization:** Used symptom screening template with three related symptom options |
| 5 | | **Patient:** Yes, some chest tightness, and difficulty with deep breathing. | 【State Update 2】 • **Evidence Set E:** E ∪ {chest tightness, breathing difficulty} • **Probability Recalculation:** - P(Bacterial Bronchitis\|E) = 75% • **Risk Assessment:** Respiratory warning symptoms detected • **Decision:** Still require symptom duration confirmation for risk stratification |
| 6 | | **Doctor:** Thank you, this helps. **Did this breathing difficulty start just today?** | 【Generate Query Q3】 • **Objective:** Obtain symptom duration • **Temporal Confirmation:** Designed specific question targeting symptom timing characteristics |
| 7 | | **Patient:** No, it began yesterday afternoon. | 【Final Completion】 • **Evidence Set E:** E ∪ {breathing difficulty(duration~1.5d)} • **Final Probability:** - P(Bacterial Bronchitis\|E) = 85% • **Confidence Validation:** Bootstrap sampling verification, 95%CI [82%, 88%] • **Decision:** Terminate consultation cycle |
| 8 | | **Doctor:** Based on your persistent cough, yellow-green sputum, fever, fatigue, and breathing difficulty (lasting one day), **bacterial bronchitis is highly likely (85%).** Recommend hospital visit today, as antibiotics may be needed. | 【Generate Final Output】 • **Evidence Integration:** Map complete evidence chain to diagnostic conclusion • **Recommendation Generation:** Generate tiered medical advice based on risk assessment level |

Figure 5: The comparison of Diagnostic Reasoning Processes Between COTCAgent and DeepSeek-V3

## C.7 Baseline Model Prompt Configuration and QA Evaluation Protocol

To ensure a fair and consistent evaluation across all baseline models, we implemented a unified prompt structure for all experiments. The prompt template was carefully designed to be model-agnostic while maintaining clinical relevance and task clarity.

Unified Prompt Structure: All models received the same core prompt structure containing:

- System Role Definition: "You are a clinical diagnostic assistant that analyzes patient symptoms and medical indicators to identify potential diseases."
- Task Instructions: Clear specification of the output format requirement (disease probabilities or confidence scores)
- Context Window: Fixed-length medical context including patient demographics, symptom history, and laboratory findings
- Response Constraints: Requirements for probabilistic reasoning and uncertainty calibration

QA Evaluation Prompt Template: For quantitative accuracy assessment, we employed the following standardized QA prompt:

"Based on the following clinical case presentation, provide a ranked list of the top-3 most likely
diseases with corresponding confidence scores (summing to 1.0). Justify each selection with key supporting symptoms and clinical indicators from the case.

Patient Case:
[INSERT COMPLETE PATIENT CASE DESCRIPTION HERE]

Required Response Format:
1. Disease Name: [Probability] - [Brief Justification]
2. Disease Name: [Probability] - [Brief Justification]
3. Disease Name: [Probability] - [Brief Justification]

Total confidence must equal 1.00. Focus on clinical evidence from the case above."

Fairness Assurance Measures:

1. Prompt Uniformity: Identical prompt templates were used across all baseline models without architecture-specific optimizations
2. Context Length Normalization: All models received context windows truncated/padded to identical token lengths
3. Temperature Settings: Deterministic inference (temperature=0) for reproducible accuracy measurements
4. Post-processing Standardization: Uniform parsing and normalization of model outputs for comparative analysis

