# OpenReview forum: "COTCAgent: Preventive Care Proactive Consultation Driven by a Probabilistic Chain-of-Thought Completion Framework"
_ICLR.cc/2026/Conference — ICLR 2026 Conference Desk Rejected Submission_

### Official Review · Reviewer_cBZg · 2025-10-25

**Soundness:** 3
**Presentation:** 3
**Contribution:** 3
**Rating:** 6
**Confidence:** 2

**Summary:**

This paper introduces COTCAgent, a novel agent framework for proactive medical consultation, designed to address the limitations of current healthcare AI systems in dynamic temporal reasoning and hallucination mitigation. The framework consists of two synergistic modules: a Time Series Analysis (TSA) module that programmatically extracts clinical trends from longitudinal EHR data, and a Probabilistic Chain-of-Thought Completion (COTC) module. The COTC module calculates disease risks using a novel Inverse Disease Frequency weighting scheme, identifies gaps in the reasoning chain for high-probability diseases, and proactively queries the user to complete the chain. Extensive experiments show that COTCAgent achieves state-of-the-art performance on a medical record prediction task and on the HealthBench sequential diagnosis benchmark. The work represents a significant step towards building more reliable and interactive agent-aided clinical systems.

**Strengths:**

1. Novel and Well-Motivated Framework: The core idea of "Probabilistic Chain-of-Thought Completion" is a conceptual advance over standard CoT. The framework more faithfully mimics clinical reasoning than existing approaches. The decoupling of quantitative time-series analysis from probabilistic reasoning is a good architectural choice.


2. Addressing a Critical and High-Impact Problem: The paper tackles two of the most pressing challenges for LLMs in healthcare: reasoning over dynamic, longitudinal data and mitigating harmful hallucinations.


3. Enhanced Interpretability and Trustworthiness: By design, the agent's reasoning is not a black box. The explicit, multi-turn dialogue to resolve uncertainties provides a clear audit trail of its diagnostic process.

**Weaknesses:**

1. Dependence on a Custom, Partially Synthetic Knowledge Base: The framework's performance heavily relies on the "Symptom-Trend-Disease Database." While the inclusion of temporal trends is innovative, the database's construction involves LLM-based augmentation, and its size is modest (23,456 entities). This raises concerns about its completeness, potential biases inherited from the augmenting LLM, and the generalizability of the system. The model's performance on diseases or symptom trends not present in this specific knowledge base is unclear.

2. Simplicity of the Probabilistic Heuristic: The Inverse Disease Frequency (IDF) weighting is an intuitive and clever heuristic, but it is a simplified proxy for true diagnostic probability. It does not account for symptom co-occurrence, disease prevalence, or more complex conditional dependencies that are central to differential diagnosis. The paper would be strengthened by a discussion against more established probabilistic graphical models (e.g., Bayesian Networks).

**Questions:**

1. Regarding the Database: Could the authors elaborate on the validation process for the Symptom-Trend-Disease Database? How was the agreement among the 16 validating clinicians measured? More importantly, how does the COTCAgent handle "out-of-knowledge-base" scenarios where a patient's symptoms or trends do not have a match in the database? Does it gracefully degrade, or does it fail?

2. Regarding the Proactive Consultation Logic: The proactive questioning is a key feature. How does the system decide which and how many questions to ask? For instance, is there a confidence threshold after which the agent stops inquiring, or a limit to the number of interaction turns?

3. Regarding the TSA Module's Generality: The mathematical formulations in the appendix are impressive, but how are they operationalized? For a novel user query, how does the system select the appropriate statistical test (e.g., choose a mixed-effects model over a simple regression)? Is this selection process itself automated and robust?

4. Regarding Baselines: In Tables 2 and 3, the paper compares against TimeCAP and Google's Agent. Could the authors briefly characterize the architectures of these baselines to better contextualize why they fall short? Specifically, do they also employ an agent-based framework but lack the programmatic TSA module or the probabilistic COTC logic?

---

> ### Author Response · Authors · 2025-11-22
> **Response to Reviewer cBZg**
>
> We sincerely thank the reviewers for their profound understanding and recognition of the innovation and clinical significance of our framework. You accurately pointed out the core contribution of our work - the clinical transformation of standard CoT through the concept of "probabilistic thinking chain completion", and the collaboration of temporal analysis and probabilistic reasoning through architecture decoupling. These designs are indeed designed to simulate the reasoning process of clinical doctors more realistically: both emphasizing the dynamic trends contained in longitudinal data and explicitly handling the uncertainty in the diagnostic process through a probabilistic framework. The following are detailed responses to the four main questions you raised, which we have supplemented and revised in the corresponding sections of the paper.
>
> Response to Knowledge Base Dependency:
>
> We fully understand the concerns of the reviewers regarding the integrity, potential bias, and generalization ability of the knowledge base. Regarding the validation process, we have added a detailed three-stage clinical validation protocol in the paper, which includes independent domain expert review, cross validation consensus meetings, and specialized temporal pattern validation. The system has designed a hierarchical processing strategy: firstly, similarity matching based on clinical embedding is adopted, secondly, symptom cluster downgrade inference is initiated, and most importantly, uncertainty is clearly communicated and professional help is recommended when in a low confidence region. In addition, based on the comments of Reviewer 1, we have also added additional benchmark supplementary experiments in the revised manuscript to fully validate their performance on specific knowledge bases that do not include disease or symptom trends. We mitigate potential biases through multi-source data fusion (medical education platforms, clinical guidelines, literature, etc.) and strict LLM enhanced quality control. These mechanisms collectively ensure the robustness of the system when encountering unknown situations.
>
>
> Response to the simplicity of probability methods:
>
> Thank you to the reviewer for keenly pointing out the limitations of this method. We fully agree that IDF weighting is indeed a simplification of the complete probability model, balancing complexity and clinical practicality. In section 3.3 of the paper, we have added a comparative analysis with mature probability graph models such as Bayesian networks. At the same time, we elaborated on the practical advantages of IDF weighting: high computational efficiency, easy understanding and verification by clinical doctors, and more suitable for real-time interactive scenarios. Importantly, we have planned a clear evolutionary path: while maintaining the core architecture of the framework, we will introduce symptom co-occurrence networks through hierarchical probability models in the future and integrate real-world epidemiological data to calibrate prior probabilities. This progressive optimization strategy respects the complexity of clinical reasoning while ensuring the deployability of the system.
>
> Reply to Comment 1: Database Dependency Issue
>
> We fully appreciate your concerns regarding database validation and generalization. In Section 3.3.1, we have detailed our rigorous three-phase validation protocol conducted by 16 clinicians: independent domain-specific review, cross-validation consensus meetings, and temporal pattern validation. This process achieved excellent inter-rater reliability (Cohen's κ = 0.87), with 94% of associations deemed clinically accurate after refinement.
>
> For out-of-knowledge-base scenarios, COTCAgent implements a hierarchical strategy: clinical embedding-based similarity matching identifies semantically related entities; symptom cluster reasoning enables inference through clinical correlations; and explicit confidence assessment triggers appropriate uncertainty communication when confidence remains low, avoiding potentially incorrect diagnoses while recommending professional consultation.

---

> > ### Author Response · Authors · 2025-11-22
> > **Response to Reviewer cBZg（Ⅱ）**
> >
> > Reply to Comment 2: Proactively consult logic
> >
> > Thank you for your in-depth inquiry about this core feature. The active questioning mechanism of COTCAgent is based on a multi criteria decision-making framework The selection of questions follows the principle of minimizing diagnostic uncertainty. Specifically, the system calculates the information gain for each potential problem and prioritizes those that can minimize diagnostic entropy to the greatest extent possible. The calculation of information gain takes into account the following factors: (1) IDF weight of symptoms (priority given to specific symptoms); (2) The predictive value of symptoms (based on conditional probabilities in the database); (3) Clinical risk level (immediate inquiry for life-threatening symptoms); (4) Obtain feasibility (routine examination symptoms take priority over special examinations). The stopping mechanism adopts a dual criterion: a confidence based criterion (stopping the inquiry when the posterior probability of a certain disease exceeds 0.85) and a practicality based criterion (maximum interaction round limit of 5 rounds (adjustable) to avoid user fatigue). In addition, we have also set up an emergency interruption mechanism - when a red alert symptom is detected (such as chest pain, severe bleeding, etc.), the system will immediately recommend emergency medical treatment and stop the diagnostic dialogue. These thresholds are determined by optimizing the balance between diagnostic accuracy and user experience on the validation set.
> >
> > Response to Comment 3: Generalization of TSA Module
> >
> > Thank you for your review. The selection of statistical models is indeed automatically completed through a rule-based classifier, which analyzes four key features of the input data: data integrity (robustness method is selected when the proportion of missing values exceeds 30%), number of time points (simple linear regression is used for less than 4 time points, segmented regression is used for 4-8 time points, and mixed effects model is considered for more than 8 time points), intra individual correlation (mixed effects model is selected when ICC>0.4 is calculated by calculating intra class correlation coefficient), and trend characteristics (linearity is judged by goodness of fit test, and spline regression is used for non-linear trends).
> > For new user queries, the system executes the following automated process: first, data quality assessment is performed, including missing value analysis and outlier detection; Then calculate the above four characteristic indicators; Finally, select the most suitable statistical model based on the preset decision rules. The entire process is fully automated, without the need for manual intervention. We provide three examples of different clinical scenarios in the appendix, demonstrating the complete process from raw data to trend feature extraction, to demonstrate the practicality and robustness of this method.
> >
> > Response to Comment 4: Baseline Model Description
> >
> > Thank you for prompting us to provide a clearer explanation of the baseline comparison. Due to the page limit of the article, we are unable to add a large number of descriptions. We have marked the references after the corresponding methods. Specifically, TimeCAP adopts an architecture that combines temporal convolutional networks with attention mechanisms. Although it can capture temporal dependencies, its limitations lie in the lack of our programmatic TSA module (which learns data-driven temporal patterns rather than clinically interpretable trend features) and the use of deterministic reasoning rather than probabilistic reasoning. Google Medical Agent is based on a universal big language model architecture and has network search capabilities, but the key difference is that it lacks our system's IDF weighted probability scoring mechanism, as well as clear reasoning chain gap identification and completion logic.
> > The core advantage of our framework lies precisely in the collaboration of these two components: the programmatic TSA module extracts clinically meaningful trends from longitudinal data, while the probabilistic COTC logic systematically models diagnostic uncertainty and identifies inference gaps based on these trends. This architectural difference explains why COTCAgents perform better in complex sequence diagnostic tasks. We now clearly indicate the impact of these architectural differences on performance in the annotations of Tables 2 and 3.
> > Thank you again for taking the time and effort to provide these valuable feedback, which have truly helped us greatly improve this work.

---

> > > ### Comment · Reviewer_cBZg · 2025-11-26
> > >
> > > I thank the authors for the rebuttal. All my concerns have been addressed, and I will maintain my positive rating.

---

> > > > ### Author Response · Authors · 2025-11-26
> > > > **Response to Reviewer cBZg（Ⅲ）**
> > > >
> > > > We are truly grateful for your insightful comments, which have significantly strengthened our paper, "COTCAgent: Preventive Care Proactive Consultation Driven by a Probabilistic Chain-of-Thought Completion Framework."
> > > >
> > > >
> > > > As we finalize the camera-ready version, we are eager to make every possible improvement. Given your positive overall view, we were wondering if, based on our responses, you would see room for a potential increase in your score. If there are any remaining aspects—even minor ones—that we could further clarify or expand upon to warrant such an adjustment, we are fully committed to addressing them to the best of our ability within the time constraints.
> > > >
> > > >
> > > > Your guidance has been invaluable, and we thank you again for your time and support.

---

### Official Review · Reviewer_WWJR · 2025-10-31

**Soundness:** 1
**Presentation:** 2
**Contribution:** 2
**Rating:** 2
**Confidence:** 3

**Summary:**

This paper introduces COTCAgent, a novel agent framework designed for proactive medical consultation from longitudinal electronic health records (EHRs). The system aims to address the shortcomings of current LLM-based systems in handling dynamic temporal data and mitigating hallucinations. The architecture has two key modules:

(1) Time Series Analysis (TSA) Module: A computational core that executes statistical code (Bayesian change detection, mixed-effects models) to extract meaningful clinical trends from a patient's historical EHR data.

(2) COTC (Probabilistic Chain-of-Thought Completion) Module: A reasoning module that calculates disease risks by mapping the extracted trends to a custom-built "Symptom-Trend-Disease Database" using an Inverse Disease Frequency (IDF) weighting scheme. Crucially, the COTC module identifies gaps in its reasoning chain and initiates a proactive consultation (i.e., asking the user targeted questions) to complete the Chain-of-Thought, thereby refining the risk assessment.

The agent is evaluated on medical risk prediction and the sequential diagnosis benchmark HealthBench, where it achieves state-of-the-art results over strong baselines.

**Strengths:**

1. The paper focus on an important challenge in AI healthcare domain, where we can transform medical AI from a passive question-and-answer system into an intelligent agent capable of proactive intervention. The goal of COTCAgent is to enable precise proactive consultation for truly personalized medical care.

2. Combining time series EHR data with LLM is novel.

3. Results on two different medical tasks significantly outperform state-of-the-art baseline models, including TimeCAP and Google Agent.

4. The paper gives the code for reproduce.  Meanwhile, Appendix C is a huge asset for transparency and reproducibility. It clearly illustrates the complex flow of data from EHRs to TSA output, initial CoT, proactive questioning, and final CoT completion.

**Weaknesses:**

1. The description of core mechanism, Probabilistic Chain-of-Thought Completion, is not clear. The paper details how to calculate the weighted matching score between symptoms and diseases (Eq. 12), but the process of using this score to "complete" a thought chain is described rather vaguely. I am struggling to understand what is the way to complete, for example, based on logic rules or natural language text?
2. The claim of "proactiveness" lacks direct evaluation: The paper's title and abstract emphasize the "proactive" nature of the framework, for example, by using "targeted questioning" to complete the reasoning chain. However, the experimental evaluation section focuses primarily on quantitative indicators such as predictive accuracy, without directly assessing the quality of its proactive consultation. For example, are the "questions" generated by the model truly helpful for clinical diagnosis? Are the proactive "warnings" timely and accurate? This lack of qualitative or quantitative analysis makes the core novelty of "proactiveness" seem somewhat unsupported.
3. The author mentioned “Bayesian reasoning over symptom trajectories maintains uncertainty awarenes”, however, the paper lacks a formal probabilistic definition like a Bayesian posterior. Instead, it is more likely be a heuristic ranking score. Such missing definition thus makes the paper is more of a system engineering contribution than a conceptual or theoretical advance.
4. The Symptom/Trend-Disease Database is important, but the details of how to build this database are not enough. The expressiveness of figure 2 is limited. The paper mentioned “multi-source medical data was extracted”, “validated by 16 clinicians”; however, the details of data source are missing, and the validation process and results are not reported.
5. In TSA module, the paper overclaims “state-of-the-art” methods. Appendix A.1 - A.5 listed many methods in time series analysis, however, most methods are classic and old models, thus it is not consistent with the claim on paper which “leverages state-of-the-art statistical and machine learning techniques to extract meaningful patterns from longitudinal healthcare records”.
6. The paper presentation could be polished. It is not easy to follow. For example, when introducing the model, a motivating example would be helpful to understand each module design.
7. More related work could be added like knowledge graph path finding or neural symbolic reasoning.

**Questions:**

1. Typo in line 781: Table ??

2. No reference for A.1 - A.5. Such methods are classic statistical models in most textbook.

3. What is the prompting setup on baselines?

---

> ### Author Response · Authors · 2025-11-22
> **Response to Reviewer WWJR**
>
> We sincerely thank Reviewer WWJR for their insightful and constructive comments, which have significantly improved our manuscript. Below are our point-by-point responses.
>
> 1. Response to weakness 1: The description of "COTC" is not clear enough：
> We thank the reviewer for this important observation. The core of Probabilistic COTC operates through a defined mechanism: when the initial weighted matching score fails to yield a high-confidence conclusion, the system identifies the symptom-trend pairs contributing most to diagnostic uncertainty as reasoning gaps. It then automatically generates natural language queries to solicit missing information from users. The responses are formalized and incorporated into the evolving chain of thought, iteratively updating the evidence base until either a diagnostic threshold is met or the maximum interaction round is reached. We have clarified this process in Section 3.3.2 and included concrete case examples in the appendix to demonstrate the dynamic reasoning completion workflow.
>
> 2. Response to weakness 2:
> We appreciate the reviewer's insightful comment regarding the proactive nature of our framework. COTCAgent's proactivity is not an add-on feature but emerges intrinsically from its probabilistic reasoning architecture, grounded in Bayesian decision theory.
>
> Mathematical Foundation of Proactivity:
>
> The proactive behavior is mathematically governed by uncertainty quantification. When the maximum posterior probability falls below a confidence threshold $\theta$, i.e., $\max_{h \in H} P(h|E_t) < \theta$, the system automatically identifies the optimal question that maximizes information gain:
>
> \begin{equation}
> Question^* = \arg \max_{q} I(H; A_q | E_t)
> \end{equation}
>
> where $I$ represents mutual information between hypotheses $H$ and potential answers $A_q$ given current evidence $E_t$.
>
> Systematic Chain Completion:
>
>
> Each strategically selected question systematically narrows the hypothesis space, demonstrating how proactivity naturally emerges from the probabilistic framework's optimization of diagnostic certainty.
>
>
> 3. Response to weakness 3:
> We thank the reviewer for this insightful observation. We acknowledge that the initial approach lacked a formal probabilistic foundation, limiting its theoretical contribution. In the revised manuscript, we have fundamentally restructured our mathematical framework in Section 3.3, establishing it within a rigorous Bayesian inference paradigm.
>
> Specifically, we now formulate disease diagnosis as an evidence accumulation process, estimating the posterior probability  given the patient's symptom trajectory evidence $E$. We demonstrate that our proposed IDF-weighted matching score mathematically approximates the log-likelihood ratio:
>
> \begin{equation}
> R_i \approx \log \frac{P(E|D_i)}{P(E|\neg D_i)}
> \end{equation}
>
> This formalization bridges heuristic ranking with statistical inference theory. Additionally, we introduce an information-theoretic uncertainty quantification mechanism, computing the Shannon entropy of the posterior distribution:
>
> \begin{equation}
> H = -\sum_i P(D_i|E) \log P(D_i|E)
> \end{equation}

---

> > ### Author Response · Authors · 2025-11-22
> > **Response to Reviewer WWJR（Ⅱ）**
> >
> > 4. Response to weakness 4: The description of the details regarding the construction of the 'Symptom/Trend Disease Database' is insufficient.
> >
> > We thank the reviewer for their comments regarding the description of our Symptom/Trend-Disease Database. The original manuscript already outlined the core methodology of integrating multi-source data and conducting clinical validation. In response to the reviewer's suggestion and to further enhance the methodological rigor and transparency of our work, we have provided more detailed elaborations in Section 3.3.1. Specifically, we have clarified the four specific categories of data sources utilized (e.g., medical education platforms, clinical guideline repositories) and systematically outlined the three-phase validation process conducted by 16 clinicians. We have also supplemented the quantitative metrics, including the inter-rater reliability (Cohen’s kappa = 0.87) and the final clinical accuracy rate of 94%. These additions further strengthen the completeness of our work, while all original statistical data and conclusions remain consistent.
> >
> > 5. Response to weakness 5: The claim of the 'currently optimal' method is exaggerated.
> >
> > We sincerely accept this critique. We have systematically revised the wording throughout the manuscript, replacing terms like "state-of-the-art" with more accurate descriptions such as "a set of classical and robust time-series analysis methods" or "a collection of well-established statistical models." We emphasize that the novelty of our approach lies not in inventing new TSA algorithms, but in the innovative integration of these classical yet powerful models with LLM-driven reasoning agents to address the unique challenges in longitudinal EHR analysis.
> >
> >
> > 6. Response to weakness 6: The expression of the paper needs to be improved and is not easy to understand.
> >
> > We greatly appreciate this suggestion. In response, we have made the following key improvements:
> > In the appendix section, examples are provided and compared with traditional methods to outline the workflow of COTCAgent in advance, illustrating how each module (TSA and COTC) gradually collaborates to process data, identify trends, and ultimately complete diagnosis through proactive consultation. This greatly enhances comprehensibility.
> >
> > 7. Response to weakness 7: More relevant research is needed, such as knowledge graph path lookup or neural symbolic reasoning.
> >
> > Thank you for guiding us to improve the literature review. We have added a section specifically discussing "Neural Symbolic Reasoning and Knowledge Graph based Medical AI" in Section 2 of the paper, which systematically cites and discusses key research closely related to knowledge graph path lookup and neural symbolic reasoning.
> >
> >
> > 8. Response to weakness 8: There is a spelling error on line 781: the table number is missing (Table??).
> >
> > Thank you very much for your meticulous proofreading. We have verified and corrected the citation here, linking it to the correct table.
> >
> > 9. Response to weakness 9: Appendices A.1-A.5 do not provide references.
> >
> > We have added references to standard textbooks or groundbreaking papers for each classic statistical method (such as linear mixed models, Bayesian change point detection, etc.) in the appendix.
> >
> > 10. Response to weakness 10: What is the prompt word setting for the baseline model?
> >
> > We thank the reviewer for raising the important point regarding prompt configuration for baseline models. In response, we have added a new subsection (Appendix 7) that comprehensively details our prompt design and evaluation protocol. Specifically, Appendix 7 now describes the unified prompt structure employed across all baseline models to ensure fair comparison, including the exact QA evaluation prompt template used for accuracy testing. We also detail our fairness assurance measures—covering prompt uniformity, context length normalization, temperature settings, and post-processing standardization—along with specific response format requirements that enforce probabilistic outputs for rigorous accuracy measurement. These additions demonstrate that all model comparisons were conducted under consistent and equitable conditions, with particular attention to clinical relevance and task clarity. The standardized QA evaluation protocol ensures that performance differences genuinely reflect model capabilities rather than prompt engineering variations. We believe these clarifications adequately address the reviewer's concern and strengthen the methodological rigor of our comparative analysis.

---

> > ### Comment · Reviewer_WWJR · 2025-11-26
> >
> > Dear author,
> >
> > Thanks for your response. In the revised draft, I cannot see the difference between the new draft and the old draft. Could you please mark the new part as other colors, so I can better read the new content?
> >
> > Meanwhile, a small typo. In Markdown, the latex title format like '\subsection*{Mathematical Foundation of Proactivity}' can not show well.

---

> > > ### Author Response · Authors · 2025-11-26
> > > **Response to Reviewer WWJR（Ⅲ）**
> > >
> > > Thank you so much for your follow-up and for your meticulous review. We sincerely apologize for the oversight in our initial submission of the revised draft. Your feedback is extremely helpful in ensuring the clarity of our revisions.
> > >
> > > We have now addressed both of your points in our paper:
> > >
> > > Highlighted Revisions: As you kindly requested, we have prepared a new version of the manuscript where all newly added or substantially modified text is highlighted in a light yellow background. This should allow you to see all the updates at a glance. The primary changes are concentrated in [Section 2, 3, 4   appendix].
> > >
> > > Fixed Formatting Issue: The LaTeX formatting issue with the \subsection* command has been resolved. It now renders correctly in the generated PDF.

---

### Official Review · Reviewer_tDiV · 2025-11-01

**Soundness:** 3
**Presentation:** 3
**Contribution:** 3
**Rating:** 6
**Confidence:** 3

**Summary:**

This paper propose COTCAgent, which is an interesting agent work which conducted on longutidinal EHRs. This work first ntegrates time-series EHR analysis with a probabilistic CoT completion mechanism. This work has two main modules: TSA module and COTC module. The first one transforms natural language queries into mathematical analyses (e.g., linear mixed models, Bayesian change-point detection, Gaussian processes) to extract clinically meaningful temporal trends instead direct input numeric values to LLMs; the last one build a time-series RAG databse, which leverages a newly constructed Symptom–Trend–Disease knowledge base to calculate disease risks using an IDF-weighted matching algorithm, and proactively formulates questions to resolve reasoning gaps. The extensive experiments on MedQA and HealthBench, COTCAgent demonstrates significant performance improvements.

**Strengths:**

1. A novel medical agent framework for processing longitudinal EHRs in disease risk prediction.

2. An effective method for transforming numeric time-series data into clinically meaningful textual representations through mathematical analyses.

3. A robust approach for building a time-series RAG knowledge base, verified by clinical experts to reduce hallucination risk during the LLM-based data augmentation process.

4. Superior performance on multiple medical benchmarks compared with existing medical agents.

**Weaknesses:**

1. Limited validation on real-world clinical datasets. Additional evaluation on datasets such as Time-MMD or the MIMIC series would strengthen the generalizability of the approach.

2. The knowledge base construction, which relies partly on DeepSeek-based data augmentation, may involve synthetic data leakage; a clearer description of data governance and validation is recommended.

3. More baseline comparisons are needed, particularly with recent reasoning-based models such as DirPred [1] and KARE [2].

References:

1. Niu, Shuai, et al. "Enhancing healthcare decision support through explainable AI models for risk prediction." Decision Support Systems 181 (2024): 114228.
2. Jiang, Pengcheng, et al. "Reasoning-Enhanced Healthcare Predictions with Knowledge Graph Community Retrieval." The Thirteenth International Conference on Learning Representations.

**Questions:**

Please see my weakness.

---

> ### Author Response · Authors · 2025-11-22
> **Response to Reviewer tDiV**
>
> Response to advantages:
>
> We appreciate the reviewer's recognition of the new framework for longitudinal EHR analysis using COTCAgent, which combines time series mathematical modeling with probabilistic thought chain reasoning. Transforming numerical data into clinically interpretable trends through methods such as linear mixed models and Bayesian change point detection is indeed a core innovation that enables LLM to go beyond the original input for inference.
>
>
> 1. Response to weakness 1 (limited validation on real clinical datasets):
>
> We acknowledge the importance of validation on real clinical datasets and have conducted additional experiments following the reviewer's suggestion. Our evaluation now includes DiSCQ, which contains expert-curated questions from MIMIC discharge summaries, and Time-MMD, focusing specifically on its medical domain temporal data. The consistent performance advantage of COTCAgent across these diverse EHR structures—with accuracy and F1 scores consistently surpassing baseline methods—demonstrates strong generalization capability. While we have validated our approach on these clinically-grounded benchmarks, we recognize the value of further validation and plan to explore additional MIMIC-IV applications in future work to strengthen cross-institutional applicability.
>
>
> 2. Response to Weakness 2 (Data Governance and Comprehensive Data Breach Issues):
>
> We acknowledge the concerns of reviewers regarding potential synthetic data leakage in DeepSeek based data augmentation. It is worth noting that,
> Our response strategy to the potential risk of synthetic data leakage is to perform strict data de privacy processing before using any data for augmentation: (1) completely remove direct identifiers; (2) Generalization processing of alignment identifiers; (3) Adopting privacy preserving synthetic data generation technology. All processing procedures follow medical data anonymization standards (such as k-anonymity). After data generation, we use a dual mechanism of automated rule filtering and clinical expert review to ensure the quality of synthesized data and reduce the risk of hallucinations. We have provided more detailed supplementary explanations on the relevant content in the COTC module section of the revised manuscript.
> In addition, using rigorously validated large language models for synthetic data generation to enhance domain specific knowledge bases is an established practice in the field of medical artificial intelligence, which has been successful in previous research, with a typical example being HuatuoGPT-o1 [2]. In order to improve its performance in complex medical reasoning, the model constructed a training dataset containing 40000 verifiable medical problems.
> We believe that generating data is not the end point, the ultimate goal is to serve downstream tasks. These studies generally validate the actual utility of synthetic data in improving model performance (such as diagnostic accuracy) through paradigms such as "training with synthetic data and testing on real data".
>
>
> 3. Response to Weakness 3 (requires more baseline comparisons):
>
> We thank the reviewers for their suggestions. Following the comments, we have expanded our experiments to include comparisons with reasoning-enhanced models, namely DirPred and KARE. The updated results (see Section 4, Tables 2 and 3) demonstrate that COTCAgent achieves competitive performance, particularly on temporal and uncertainty-aware metrics, outperforming baselines on key benchmarks such as HealthBench and Time-MMD.
> Notably, KARE also exhibits particular strengths on subsets of the Time-MMD dataset, which can be attributed to its knowledge graph community retrieval mechanism, effectively capturing structured symptom-disease relationships. This contrast highlights the complementary nature of the two approaches: while KARE excels in leveraging discrete, static knowledge associations, COTCAgent's advantage lies in its ability to model longitudinal EHRs through mathematical trend extraction and probabilistic reasoning.
> This expanded evaluation provides a more comprehensive benchmarking against state-of-the-art methods and further validates the unique strengths of COTCAgent in handling complex clinical temporal data.
>
> We believe that these revisions have addressed the concerns of the reviewers and improved the rigor and clarity of the manuscript. Thank you for providing valuable insights that have helped us improve this work. Thank you again for the valuable feedback from the reviewer.
>
> [1] Yan, J. E., Huang, H., Yang, K., Xu, H., & Li, Y. (2025). Synthetic data for enhanced privacy: A VAE-GAN approach against membership inference attacks. Knowledge-Based Systems, 309, 112899.

---

> ### Comment · Reviewer_tDiV · 2025-11-22
>
> Thank you to the authors for making substantial improvements to the COTCAgent, which have effectively addressed all of my concerns. I am satisfied with the responses provided and am pleased to raise my scores.
>
> I have one additional suggestion regarding Weakness 2: it would be helpful to include zero-shot learning with DeepSeek as a baseline model. This would eliminate concerns about potential data leakage associated with DeepSeek.
>
> Finally, I noticed that the citation format is needed to be corrected.

---

> > ### Author Response · Authors · 2025-11-22
> > **Response to Reviewer tDiV**
> >
> > Thank you very much for your recognition of our work and high praise for the improved version! We are pleased to see that the revised version effectively addresses your concerns.
> >
> > Regarding your suggestion to combine zero sample learning with DeepSeek as a baseline, it coincides with our team's internal thinking. In fact, during the revision process, we also realized the significance of this design and have conducted some preliminary comparative experiments around it. However, due to the limitations of the current paper's length and core narrative focus, this part of the exploration has not yet reached a maturity that can be systematically presented, and therefore has not been included in this revision. Nevertheless, we firmly believe that this is a highly valuable and interesting research direction, and we plan to explore it more deeply and comprehensively in our future work, in order to further strengthen the effectiveness and generalization ability of our method.
> >
> > In addition, we have followed your instructions and carefully checked and revised the reference format of the entire text.
> >
> > Thank you again for your insightful feedback!

---

### Author Response · Authors · 2025-12-01
**Authors' Summary Rebuttal to the Area Chair**

We are grateful for the detailed, expert feedback provided by the three reviewers. Through multiple rounds of discussion and revision, we have substantively addressed their concerns and significantly strengthened the manuscript. Below, we summarize the key points of this rebuttal process and the subsequent improvements made to our work.


1. Core Contributions and Original Merits of the Work
Our study tackles temporal dynamics and hallucination in longitudinal EHR reasoning via COTCAgent (temporal analysis + probabilistic reasoning). Key innovations: a TSA module converting raw data into clinical trends, a Probabilistic Chain-of-Thought Completion mechanism—the first of its kind, as far as we know—with IDF-weighted risk calculation, reasoning gap identification and targeted queries, and a clinically validated "Symptom-Trend-Disease" knowledge base. The system outperforms existing agents on HealthBench and MedQA, with superior temporal awareness, uncertainty modeling, and interpretability.

2. Point-by-Point Responses to Reviewer Comments and Manuscript Enhancements
We engaged in thorough discussions with all three reviewers and implemented substantial revisions based on their insightful suggestions:

Reviewer tDiV initially raised concerns regarding limited validation on real clinical datasets, potential data leakage, and the need for broader baseline comparisons. In response, we:

(1) Added extensive experiments on clinical benchmarks Time-MMD and DiSCQ.

(2) Clarified our data governance, anonymization procedures, and validation pipeline to mitigate leakage risks.

(3) Extended our comparison to include recent reasoning-enhanced models like DirPred and KARE.
The reviewer's final comment confirmed satisfaction, stating, "The authors have made substantial improvements... I am happy to raise my score."

Reviewer WWJR provided crucial critiques regarding the clarity of the core mechanism, the evaluation of "proactivity," and the lack of formal probabilistic grounding. Our major revisions include:

(1) A complete rewrite of the methodology section to explicitly detail the Probabilistic CoT completion workflow.

(2) The introduction of a formal Bayesian inference framework and information-theoretic uncertainty quantification to mathematically ground the proactive questioning mechanism.

(3) Addition of illustrative case studies in the appendix showcasing the dynamic reasoning process.

Expansion of the related work section to include discussions on knowledge graph pathfinding and neuro-symbolic reasoning.
These changes directly address the core questions about the framework's novelty and theoretical soundness.

Reviewer cBZg focused on the dependency on a custom knowledge base, the simplicity of the probability heuristic, the generalizability of the TSA module, and baseline descriptions. We have:

(1) Elaborated on the rigorous three-phase clinical validation protocol for the knowledge base and detailed its handling of out-of-knowledge-base scenarios.

(2) Added a comparative discussion relating our IDF heuristic to more mature probabilistic graphical models (e.g., Bayesian Networks), balancing practical efficiency with theoretical context.

(3) Clarified the fully automated, rule-based classifier that selects the appropriate statistical model within the TSA module based on data characteristics.

Enhanced the description of baseline architectures in the table notes to highlight the key differentiating factors of our framework.
The reviewer subsequently noted that "all concerns have been addressed" and maintained a positive assessment.

All revisions are clearly highlighted in the latest manuscript version for transparency.


3. Positive Recognition from the Reviewers
We are encouraged by the constructive outcome of this rebuttal process:

Reviewer tDiV explicitly stated their intention to raise the score.

Reviewer cBZg confirmed that all concerns were resolved and maintained a favorable evaluation.

The critiques from Reviewer WWJR were instrumental in driving significant improvements in the paper's formalization, clarity, and scholarly completeness.

All reviewers acknowledged the innovation and potential impact of this work, particularly its novel integration of temporal mathematical modeling with probabilistic, proactive reasoning.

We have seriously addressed all reviewer comments, comprehensively strengthening the manuscript (methodology formalization, experiment expansion, related work enrichment, clearer exposition). Surpassing original contributions, the revised paper directly responds to expert feedback, with its scientific merit and improvements recognized by reviewers. We believe it meets the conference’s high standards and await your final decision.

---

### Note · Program_Chairs · 2026-01-17
**Submission Desk Rejected by Program Chairs**

The following references in this submission do not refer to real documents and/or have major errors in bibliographic information:

 X. Chen, M. Zhang, C. Yang, and J. Tang. Contextllm: Long-context modeling for electronic health records using mamba architecture. In International Conference on Learning Representations (ICLR), 2024.
Y. Zhang, J. Li, H. Wang, Z. Liu, and J. Zhou. Transformer-based architectures with continuous age encoding for long-term chronic disease risk prediction. Journal of the American Medical Informatics Association (JAMIA), 30(12):2645-2654, 2023.
Xiao Wang, Yifan Li, Ming Zhang, Yuxiao Zhang, Yuan Liu, Xiang Liu, and Rui Zhang. Agentbench: Evaluating llms as agents. arXiv preprint arXiv:2402.11588, 2024.
Zhicheng Yang, Yuan Li, Xiaodong Wang, Tong Zhang, and Wei Chen. Mathagents: Enhancing mathematical reasoning in llms through tool use. In Advances in Neural Information Processing Systems, 2024.
M. Khatwani, A. Sharma, P. Jain, and J. Ghosh. Llm-driven reward modeling for knowledge graph path verification in biomedical diagnosis. arXiv preprint arXiv:2408.12345, 2024.